# Dissociable control of unconditioned responses and associative fear learning by parabrachial CGRP neurons

**Anna J Bowen[1,2,3], Jane Y Chen[1,2], Y Waterlily Huang[1,2], Nathan A Baertsch[4], Sekun Park[1,2], Richard D Palmiter[1,2]***

[1]Department of Biochemistry, University of Washington, Seattle, United States; [2]Howard Hughes Medical Institute, University of Washington, Seattle, United States; [3]Graduate Program in Neuroscience, University of Washington, Seattle, United States; [4]Center for Integrative Brain Research, Seattle Children's Research Institute, Seattle, United States

**Abstract** Parabrachial CGRP neurons receive diverse threat-related signals and contribute to multiple phases of adaptive threat responses in mice, with their inactivation attenuating both unconditioned behavioral responses to somatic pain and fear-memory formation. Because CGRP[PBN] neurons respond broadly to multi-modal threats, it remains unknown how these distinct adaptive processes are individually engaged. We show that while three partially separable subsets of CGRP[PBN] neurons broadly collateralize to their respective downstream partners, individual projections accomplish distinct functions: hypothalamic and extended amygdalar projections elicit assorted unconditioned threat responses including autonomic arousal, anxiety, and freezing behavior, while thalamic and basal forebrain projections generate freezing behavior and, unexpectedly, contribute to associative fear learning. Moreover, the unconditioned responses generated by individual projections are complementary, with simultaneous activation of multiple sites driving profound freezing behavior and bradycardia that are not elicited by any individual projection. This semi-parallel, scalable connectivity schema likely contributes to flexible control of threat responses in unpredictable environments.

*For correspondence:
palmiter@u.washington.edu

## Introduction

Imminent threats such as somatic pain rapidly shape ongoing behavior and alter physiology to prioritize immediate threat remediation (*LeDoux, 2000*). This cascade of activity, which in rodents can include bouts of active escape or freezing behavior (*Blanchard and Blanchard, 1969*; *Fanselow, 1982*; *Roelofs, 2017*) and autonomic changes, including both enhanced sympathetic and parasympathetic outflow (*Fitzgerald and Teyler, 1970*; *Iwata and LeDoux, 1988*), comprise the unconditioned response. A later phase of threat response includes enhanced arousal, wariness and anxiety (*Wang et al., 2015*). In tandem to these innate adaptive responses, the aversive threat signal is transmitted to forebrain nuclei that receive convergent information about ongoing environmental stimuli, and associations are formed allowing prediction of future threats based on environmental information (*Blair et al., 2001*; *Bolles and Collier, 1976*; *LeDoux, 2000*; *Maren, 2001*; *Romanski et al., 1993*; *Tovote et al., 2015*). Upon re-exposure to pain-predictive cues (e.g. an auditory conditioning stimulus (CS)), nuclei storing the associative memory are reactivated and through downstream partners trigger responses previously hallmarks of the unconditioned response (e.g. freezing behavior and autonomic arousal) (*Goosens and Maren, 2001*; *Iwata and LeDoux, 1988*; *Maren, 2001*; *Tovote et al., 2015*). Hence, while the systems controlling unconditioned responses and associative learning have dissociable processes, they have highly convergent behavioral and

physiological readouts. Due in part to this inherently entangled arrangement, dissection of these affective processes prior to the level of the amygdala has remained elusive.

The parabrachial nucleus (PBN), located at the junction of the midbrain and pons, is implicated in relaying aversive threat information to the forebrain (*Bernard and Besson, 1988*; *Chiang et al., 2019*; *Gauriau and Bernard, 2002*). A recently identified population of neurons expressing calcitonin gene-related peptide (CGRP, encoded by the *Calca* gene) resides in the external lateral PBN and is robustly activated by threats of diverse origin (*Campos et al., 2017*; *Campos et al., 2018*; *Carter et al., 2013*; *Chen et al., 2018*), including somatic pain (*Han et al., 2015*). In addition to contributing to affective and behavioral responses to pain, CGRP[PBN] neurons are necessary for associative fear learning (*Han et al., 2015*). While neurons across the entire population appear to broadly respond to multi-modal threats (*Campos et al., 2018*), it remains possible that subpopulations are preferentially activated by distinct stimuli and project to designated partners to drive appropriate responses. The alternative extreme is that CGRP[PBN] neurons are a homogeneous population with broadly distributed projections, whose distinct phenotypes are elaborated entirely by downstream partners with activity shaped by additional sensory inputs.

We sought to disentangle the organization of CGRP[PBN] to forebrain circuitry by delineating their distribution of projections and then determining whether they originate from distinct CGRP[PBN] neuron subgroups or arise by collateralization. To interrogate the underlying logic by which unconditioned responses and associative learning are simultaneously driven from this single population, we selectively activated individual terminal fields in downstream targets and measured their individual capacity to elicit behavioral and physiological changes and/or contribute to associative fear learning. We found that many distinct phenotypes were produced by discrete projections, while a selected few contribute to associative fear learning.

## Results

### CGRP[PBN] neurons generate learned and innate defensive responses and connect to diverse forebrain targets

To determine whether activation of CGRP[PBN] neurons is sufficient to induce both the behavioral and physiological correlates of the unconditioned response in addition to fostering associative fear learning (*Han et al., 2015*), we bilaterally injected an adeno-associated virus carrying Cre-dependent channelrhodopsin (AAV1-DIO-ChR2:YFP) and implanted fiber-optic cannulae over the PBN of *Calca*[Cre/+] mice, while control mice received AAV1-DIO-YFP (*Figure 1A*; *Figure 1—figure supplement 1A*). Repeated high-frequency (30 Hz) activation of CGRP[PBN] neurons induced profound freezing behavior (*Figure 1B*, *Figure 1—figure supplement 1B–C* and *Figure 1—video 1*), as indicated by rigid, uninterrupted immobility. In addition to eliciting robust freezing behavior, we confirmed that pairing photostimulation with an auditory CS rapidly induced conditioned freezing responses to the CS (*Figure 1C*; *Han et al., 2015*). To test whether CGRP[PBN] neurons can recapitulate physiological aspects of the unconditioned response, we photostimulated the neurons while monitoring heart rate with a pulse oximeter (*Figure 1D*). Interestingly, while modest activation (15 Hz, subthreshold for eliciting freezing behavior) resulted in moderate tachycardia (*Figure 1—figure supplement 1E*), high-frequency activation (30 Hz) led to profound bradycardia and decreased respiration followed by dramatic post-stimulation rebound tachycardia and mild hyperventilation (*Figure 1E–F*, respiration measured in plethysmography chamber); it also produced vasoconstriction (*Figure 1—figure supplement 1H*; *Vianna and Carrive, 2005*). Hence, CGRP[PBN] neurons are capable of exerting opposing effects on autonomic physiology depending on their activation frequency. Comparing the latencies of somatic vs autonomic responses to 30 Hz photostimulation, we found that freezing behavior is rapidly initiated (median 0.42 s), while bradycardia takes longer to develop (median 22.15 s, *Figure 1—figure supplement 1C–D*), suggesting that freezing behavior does not emerge simply as a consequence of adverse autonomic effects. To test whether CGRP[PBN] neurons can also elicit behavioral alterations associated with late-phase responses to threat exposure, we subjected mice to an elevated-plus-maze test (*Martin, 1961*; *Pellow et al., 1985*) while activating CGRP[PBN] neurons; this treatment attenuated open-arm exploration consistent with an anxiogenic effect (*Figure 1G*).

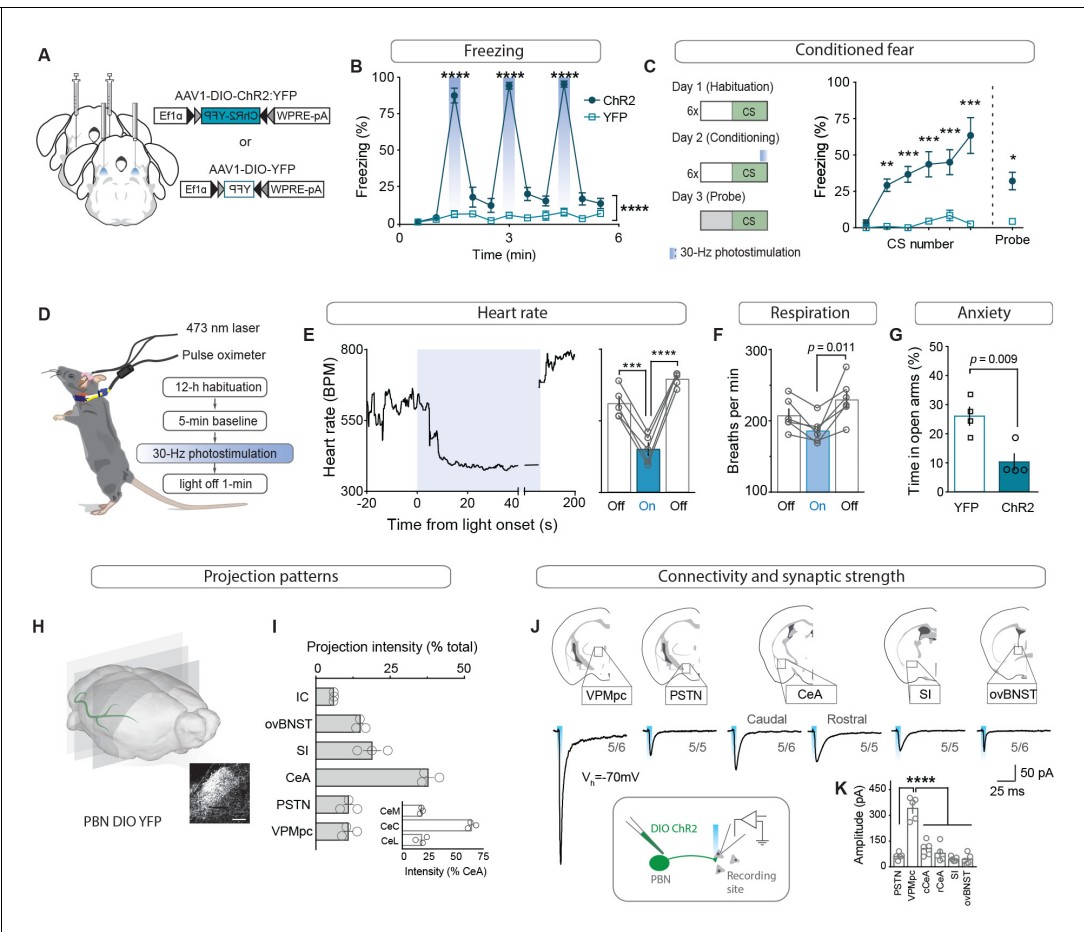

**Figure 1.** CGRP$^{PBN}$ neurons potentiate fear behavior, drive associative learning and robustly activate forebrain targets. (**A**) Bilateral injections of AAV1-DIO-ChR2:YFP or AAV1-DIO-YFP and fiberoptic cannula implants above the PBN of Calca$^{Cre/+}$ mice. (**B**) Photostimulation (30 Hz) of CGRP$^{PBN}$ neurons generated robust freezing behavior (n = 8,6 (n = ChR2, YFP); significant group x time interaction in a two-way ANOVA, $F_{10,120}$ = 83.53, p<0.0001; subsequent Sidak pairwise comparisons, ****p<0.0001). (**C**) Optogenetic stimulation of CGRP$^{PBN}$ neurons conditioned freezing behavior when preceded by a 10 kHz auditory CS (n = 4,4 (n = ChR2, YFP); significant group x time interaction in a two-way ANOVA, $F_{5,36}$ = 5.62, p=0.0006; subsequent Sidak pairwise comparisons, **p<0.01; ***p<0.001; and Welch's unpaired t-test for probe trial, $t(3.47)$ = 5.62, *p=0.016). (**D**) Schematic and timeline for pulse-oximetry measurements of autonomic responses to optogenetic stimulation. (**E**) Representative and mean bradycardia caused by 30 Hz photostimulation of CGRP$^{PBN}$ neurons (n = 5, one-way ANOVA, $F_{2,12}$ = 39.66, p<0.0001; subsequent Dunnett correction for multiple comparisons). (**F**) Respiratory rate was also reduced during photostimulation (n = 6, one-way ANOVA, $F_{2,15}$ = 5.12, p=0.0196; subsequent Dunnett correction for multiple comparisons, p=0.011). (**G**) Stimulation of CGRP$^{PBN}$ neurons was anxiogenic (n = 4,4, Welch's unpaired t-test, $t(5.93)$ = 3.78, p=0.009). (**H**) Expression of a fluorescent protein in CGRP$^{PBN}$ neurons to identify efferent projections. Scale bar: 100 μm. (**I**) Fluorescence in downstream targets relative to cumulative projection intensity; inset is fluorescence in CeA subnuclei relative to total CeA fluorescence. (**J**) Representative light-evoked EPSCs from cells downstream of CGRP$^{PBN}$ neurons; figures below traces (e.g. 5/6) indicate proportion of recorded cells that responded within each region. (**K**) Average amplitudes of EPSCs from responsive cells (5 cells for each site from four mice; 30/33 cells responded, significance for one-way ANOVA, $F_{5,24}$ = 38.75, p<0.0001; subsequent Tukey correction for multiple comparisons). Data are represented as mean ± SEM. For full statistical information see *Supplementary file 1*.

The online version of this article includes the following video and figure supplement(s) for figure 1:

**Figure supplement 1.** Fiber placement, autonomic measurements, and contralateral projection strength.

**Figure 1—video 1.** Freezing behavior generated by activating CGRP$^{PBN}$ neurons.

https://elifesciences.org/articles/59799#fig1video1

To map the forebrain connections from CGRP$^{PBN}$ neurons that underlie their wide physiological and behavioral repertoire, we sectioned the forebrain of mice expressing a fluorescent tracer (AAV1-DIO-YFP) in CGRP$^{PBN}$ neurons and identified axon terminals in various downstream sites (*Figure 1H*). Comparing individual targets to cumulative projection intensity, we found major projections to the central amygdala (CeA,~40%, primarily targeting the capsular sub-nucleus), substantia

innominata (SI,~20%), and oval sub-nucleus of the bed nucleus of stria terminalis (ovBNST,~15%), with weaker projections to the parasubthalamic nucleus, thalamus and visceral insular cortex; PSTN and VPMpc,~10% each, IC,~5% (*Figure 1I*, for abbreviations see figure supplement 2). With the exception of the IC, CGRP$^{PBN}$ neurons also target the contralateral hemisphere for all of their downstream partners, markedly to the contralateral PSTN and VPMpc, with ~75% and 50% of the ipsilateral projection intensity, respectively (*Figure 1—figure supplement 1I–J*). To confirm that downstream neurons receive monosynaptic excitation from CGRP$^{PBN}$ neurons and also to compare synaptic strength across targets, we expressed channelrhodopsin (ChR2) in CGRP$^{PBN}$ neurons and photostimulated terminals in downstream regions while recording from putative postsynaptic neurons in a slice preparation (*Figure 1J*). Interestingly, we found that while all the major downstream targets were recipients of reliable excitatory input from CGRP$^{PBN}$ neurons (*Figure 1J*, IC not tested), the VPMpc, while not receiving the strongest input based on fiber density, exhibited significantly greater excitation from terminal activation than any other recording site (*Figure 1K*).

The heterogeneity of behavioral and physiological outcomes elicited by activation of CGRP$^{PBN}$ neurons raises questions about the underlying circuit organization responsible for their generation. We envisioned several potential circuit structures underlying CGRP$^{PBN}$ neuron connectivity to the forebrain: while distributed, one-to-all connectivity involving extensive collateralization from each CGRP neuron to every target structure would be well suited for simultaneous, parallel activation of diverse regions, a one-to-one, segregated organization would better support separable generation of distinct functions via activation of designated partners. To reveal the structure underlying CGRP$^{PBN}$-neuron connectivity to the forebrain, we devised a method to selectively isolate subsets of CGRP$^{PBN}$ neurons as defined by their target-projecting behavior. By injecting AAV expressing retrogradely transported Flp-recombinase (rAAV2-retro-Flp) into a downstream site and a fluorescent tracer requiring both Cre and Flp for expression (*Fenno et al., 2014*) (AAV-Cre$_{on}$-Flp$_{on}$-YFP; Target +), or that is turned on by Cre but off by Flp (*Fenno et al., 2014*) (AAV-Cre$_{on}$-Flp$_{off}$-YFP; Target –) into the PBN of *Calca$^{Cre/+}$* mice (*Figure 2A*), we were able to isolate fluorescent expression to neuronal subpopulations defined by whether or not they targeted a region of interest (*Figure 2B*, *Figure 2—figure supplement 1A*). Normalizing the resulting projection intensity in each downstream region under each condition to the maximal signal given by transducing all CGRP$^{PBN}$ neurons, we determined the proportion of terminal density in each downstream partner supplied by target-projecting CGRP$^{PBN}$ neurons for the VPMpc, PSTN, CeA, and ovBNST (*Figure 2C*). This analysis revealed that CeA-projectors contributed substantially to PSTN, SI, VPMpc and ovBNST projections, but not IC. VPMpc-projectors, interestingly, while also projecting to the CeA, contributed more substantially to the SI and IC, while PSTN-projectors had limited secondary output to the CeA and SI, and ovBNST-projectors had only a weak secondary projection to the CeA (*Figure 2C*), shown schematically in *Figure 2E*. Quantifying the number and location of the different projecting subpopulations within the PBN revealed that neurons projecting to the CeA made up the largest proportion of CGRP$^{PBN}$ neurons residing within the external lateral PBN, while neurons projecting to VPMpc accounted for most of the CGRP$^{PBN}$ neurons residing in the medial and waist regions; neurons projecting to ovBNST, the smallest group, were restricted to the external lateral PBN (*Figure 2—figure supplement 1B–F*). Comparing projection distributions for the Target + or Target - expression conditions, we found that regardless of the downstream target used to drive expression, the CeA was the primary downstream partner in terms of projection intensity (*Figure 2D*). Excluding CeA-projecting CGRP$^{PBN}$ neurons flattened the distribution, with the ovBNST narrowly making up the largest projection contribution. As a summary statistic to directly compare the collateralization tendencies across subpopulations, we calculated a collateralization coefficient defined as the difference between projection strength for each downstream partner in the Target + and Target - conditions, for each target, where a value of 50% corresponds to half of the signal in the area of interest being supplied by target-site projectors (*Figure 2F*, *Figure 2—figure supplement 1G*). Looking at the distribution of these coefficients across secondary downstream partners for each target site, we found that VPMpc projectors had the greatest tendency to collateralize, while ovBNST projectors collateralized primarily to the CeA (*Figure 2G*, *Figure 2—figure supplement 1G*). In summary, there is extensive collateralization by CGRP$^{PBN}$ neurons with no one-to-one projections; rather, CGRP$^{PBN}$ neurons tend to distribute their projections among large groups of downstream targets, composing a one-to-many distributed projection arrangement (*Figure 2E*).

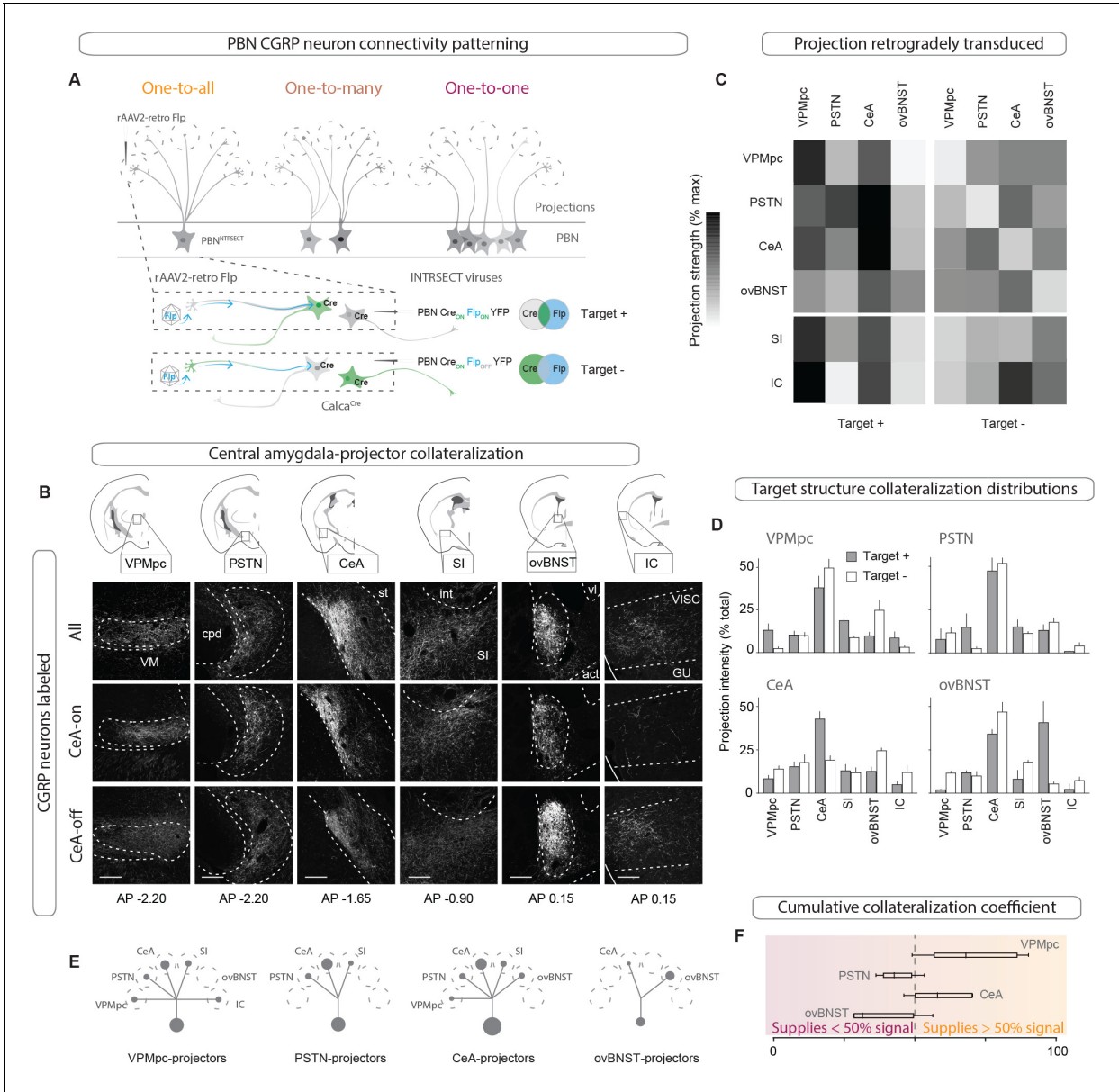

**Figure 2.** CGRP^PBN neurons broadly collateralize to forebrain targets. (**A**) Injections of rAAV2-retro-Flp into projection targets and INTRSECT viruses into the PBN of Calca^Cre/+ mice to isolate target-projecting (Target +, Cre-on Flp-on) or non-projecting (Target -, Cre-on Flp-off) populations. (**B**) Fluorescent images of projection targets in mice expressing tracer in either all CGRP^PBN neurons, CeA-projectors (CeA-on), or non-CeA-projectors (CeA-off). Scale bar: 100 μm.(**C**) Heat maps of averaged fluorescent intensity in downstream sites for Target + or Target - viral expression conditions for the VPMpc, PSTN, CeA, and ovBNST; values normalized to maximal target projection intensity given by expression of DIO-YFP (n = 3 per condition). (**D**) Overview of target-projecting projection distributions for VPMpc, PSTN, CeA, and ovBNST in Target + and Target – conditions (mean ± SEM). (**E**) Schematic of relative population size and collateralization distribution from each target-projecting subset. Collaterals were indicated if collateralization coefficient was >50% (see below), *or* if structure made up >35% projection distribution in (**D**) from Target + condition. (**F**) Collateralization coefficient calculated as difference between normalized fluorescence intensity in projection site in Flp-on condition – Flp-off condition, averaged across all sites, scaled by 50% and forced through 0 for y-intercept. Example calculation for VPMpc-projector to CeA collateralization coefficient: (([CeA fluorescence]_{VPMpc-ON} – [CeA fluorescence]_{VPMpcOFF})/[CeA fluorescence]_{DIOYFP})x 50% + 50%. Center line, mean; box limits, upper and lower quartiles; whiskers, min to max.

The online version of this article includes the following figure supplement(s) for figure 2:

**Figure supplement 1.** Collateralization to forebrain targets by CGRP^PBN neurons.

# Individual downstream targets of CGRP$^{PBN}$ neurons exert diverse effects on physiology and behavior

To assess the contribution of activating CGRP$^{PBN}$ projections to individual brain regions in eliciting behavioral and physiological processes associated with unconditioned responses to aversive stimuli, we used ChR2 to stimulate terminals within specific target regions (*Figure 3—figure supplement 1A–C*), for fiber placement summary; fibers targeting the caudal and rostral CeA were placed in the caudal or rostral third of the CeA (caudal to −1.4 AP and rostral to −1.0 AP, respectively). Because of the high degree of collateralization, it is possible that stimulating one region will result in antidromic activation and neurotransmitter release in all areas with shared innervation. If that occurred, then stimulating in one area that shares strong co-innervation with another should yield similar phenotypic outcomes. Surprisingly, given the broad collateralization of CGRP$^{PBN}$ neurons, that was not the case. Only photostimulating terminals in the VPMpc or PSTN led to reliable initiation of freezing behavior (*Figure 3A–B*,~40% time-spent freezing), while photostimulating the caudal CeA (cCeA), SI, or ovBNST had more subtle effects (~25% time-spent freezing, *Figure 3C–F*), and stimulating the rostral CeA (rCeA) actually led to a non-significant increase in locomotion (*Figure 3—figure supplement 2B*; for cross-area mean freezing response comparisons see *Figure 3—figure supplement 2A*). Notably, activating no individual projection was able to match CGRP$^{PBN}$ cell-body activation in generating robust freezing behavior (*Figure 3—figure supplement 2A,C*). Importantly, while freezing behavior is an unconditioned response to predator incursion (*Roelofs, 2017*), when elicited by noxious stimulation it is instead a learned response to contextual cues because adaptive responses to ongoing noxious stimulation are always to flee or withdraw (*Fanselow, 1982*; *Landeira-Fernandez et al., 2006*). To examine whether the freezing behavior we observed was directly elicited by photostimulation or was instead driven by processes secondary to contextual conditioning, we looked at the temporal structure of the freezing responses to light onset and offset. We found that photostimulation led to short-latency freezing bout initiation (<5 s after stimulation onset) for most terminal stimulation groups except the rCeA, which instead elicited short-latency freezing bouts after stimulation offset (~2.4 s) (*Figure 3—figure supplement 2C–D*). Freezing-bout initiation occurred with lower latencies than control animals during the 20 s post-stimulation epoch in all stimulation groups except the VPMpc and cCeA (*Figure 3—figure supplement 2C–D*). When taken together with the observation that freezing behavior occurs with greater frequency during the stimulation epoch than post-stimulation epoch for all fiber-placement groups except those in the CeA and ovBNST (*Figure 3—figure supplement 2E*), these findings suggest that stimulation of most CGRP$^{PBN}$ neuron projections simultaneously elicits a direct effect on freezing behavior while also generating aversive properties that promote transient contextual freezing. Interestingly, in the case of the rCeA, the direct effect on freezing is absent, while the contextual memory effects on freezing are instead the primary effect (*Figure 3—figure supplement 2C–E*).

By measuring the effect of photostimulating different terminal fields on multiple physiological measures, we found that activating the PSTN, rCeA, SI or ovBNST led to tachycardia, while activating the VPMpc or cCeA had no effect (*Figure 3F–L*). In addition to eliciting tachycardia, photostimulating terminals in the PSTN, rCeA or SI caused vasoconstriction (*Figure 3S–X*), while activating only the rCeA, SI, or ovBNST elicited hyperventilation (*Figure 3M–R*). Lower frequency stimulation (15 Hz) led to similar, less robust physiological effects across regions (*Figure 3—figure supplement 3A–F*), while light delivery alone in control animals had no effect on any of these measures (*Figure 3—figure supplement 3G–L*). Compellingly, the most co-innervated downstream regions – the VPMpc and SI, CeA and ovBNST, and PSTN and CeA, each had distinct effects on physiology and behavior, with some (VPMpc, PSTN) preferentially inducing freezing behavior, and others (SI, CeA, ovBNST) robustly eliciting autonomic responses, suggesting that terminal stimulation does not produce robust antidromic activation that homogeneously activates all co-innervated regions. In support of this conclusion, we observed that photostimulation of terminals in each downstream target did not generate antidromic activation of CGRP$^{PBN}$ cell bodies sufficiently to induce Fos expression in the PBN (*Figure 3—figure supplement 2G*). Taken together, these behavioral and physiological data suggest that the projections to thalamic (VPMpc) and hypothalamic (PSTN) targets elicit freezing behavior the best, while activating extended amygdalar structures (rCeA, SI, ovBNST) elicits sympathetic autonomic responses, implying a specialization in function across downstream partners.

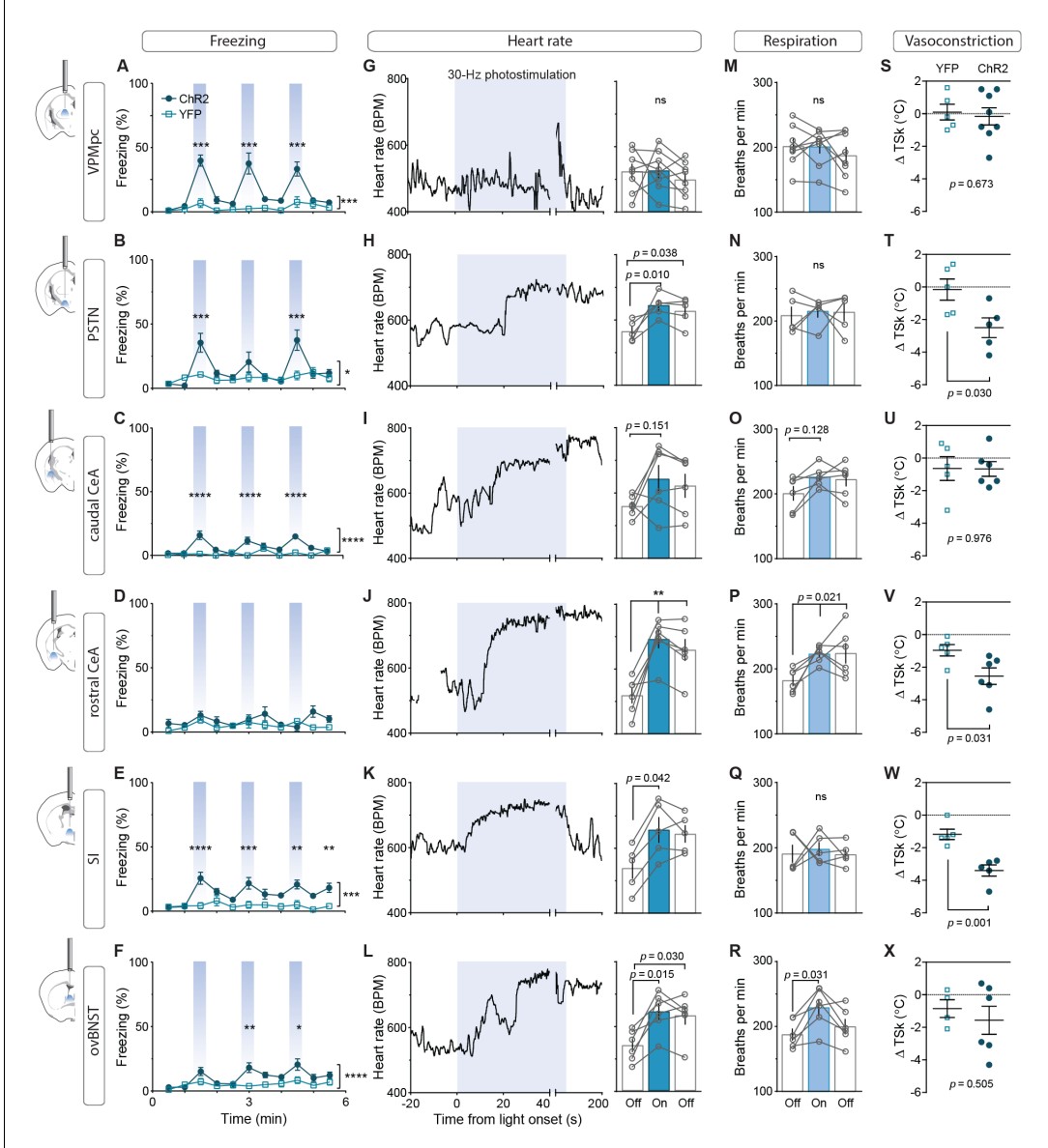

**Figure 3.** Photostimulation of CGRP[PBN] neuron terminals in individual downstream targets exerts diverse effects on physiology and behavior. (**A**) Activating terminals in the VPMpc (n = 8,5) (ChR2, YFP) elicited freezing behavior but had no effect on (**G**) heart rate (**M**) respiration or (**S**) vasoconstriction. (**B**) Photostimulating terminals in the PSTN (n = 6,5) elicited freezing behavior, (**H**) caused mild tachycardia, (**N**) had no effect on respiration but (**T**) caused vasoconstriction. (**C**) Photostimulating terminals in the cCeA (n = 6,5) increased freezing behavior but had no effect on (**I**) heart rate (**O**) respiration or (**U**) vasoconstriction. (**D**) Photostimulating terminals in the rCeA (n = 6,5) had no effect on freezing behavior (**J**) elicited robust tachycardia (**P**) hyperventilation and (**V**) vasoconstriction. (**E**) Photostimulating terminals in the SI (n = 8,6) increased freezing behavior, (**K**) caused tachycardia (n = 5), (**Q**) had no effect on respiration and (**W**) caused vasoconstriction. (**F**) Photostimulating terminals in the ovBNST (n = 9,5) increased freezing behavior, (**L**) caused tachycardia and (**R**) hyperventilation but (**X**) did not affect vasoconstriction. (**A–F**) Significance for effect of group in a two-way ANOVA with subsequent Sidak pairwise comparisons. (**G–R**) Significance for one-way ANOVA with subsequent Dunnett correction for multiple comparisons. (**S–X**) Significance for Welch's unpaired t-test. Data are represented as mean ± SEM. *p<0.05; **p<0.01; ***p<0.001; ****p<0.0001. For full statistical information see *Supplementary file 1*.

The online version of this article includes the following figure supplement(s) for figure 3:

**Figure supplement 1.** Verification of terminal stimulation of CGRP[PBN] neuron projections.

**Figure supplement 2.** Freezing behavior elicited by photostimulation of CGRP[PBN] neuron terminals.

**Figure supplement 3.** Physiological responses to photostimulation of CGRP[PBN] neuron terminals.

## CGRP$^{PBN}$-neuron downstream targets differentially influence associative learning and affect

To measure alterations in anxiety state, potentially indicative of enhanced arousal or vigilance in response to threats (*Martin, 1961*; *Mestanik et al., 2015*), we photostimulated terminals in downstream targets while mice explored an elevated-plus maze (*Figure 4A*). Only photostimulation of terminals in the ovBNST significantly reduced open-arm exploration, consistent with an anxiogenic effect, while photostimulating terminals in the rCeA paradoxically increased open-arm exploration (*Figure 4B–C*).

To further interrogate the affective state generated by activation of each downstream partner we utilized a real-time, place-preference (RTPP) assay to assess whether mice would choose to seek out or avoid terminal photostimulation (*Figure 4D*). Mice with photostimulation of either CGRP$^{PBN}$ somata or their terminals in the VPMpc, PSTN, rCeA, or SI robustly avoided photostimulation (*Figure 4E–K*, *Figure 4—figure supplement 1E–K*), whereas mice with photostimulation of terminals in the cCeA or ovBNST had no preference relative to control animals, which spent equal time in the three compartments. Considering aversive valence in combination with the observation that photostimulation of terminals in the rCeA robustly potentiated escape attempts during exposure to noxious heat (*Figure 4—figure supplement 1A–C*) without affecting spinal analgesia (*Figure 4—figure supplement 1D*), implies that activating the rCeA may not be anxiolytic per se, but shift behavior toward active coping strategies during threatening situations (*D'amour and Smith, 1941*; *Espejo and Mir, 1993*).

While we had observed that stimulation of multiple individual projections was able to transiently generate contextual freezing, we were interested in distinguishing between intrinsic effects of stimulation on freezing behavior versus secondary effects on associative learning. To accomplish this, we subjected mice to an associative fear-learning paradigm where an auditory conditioning stimulus (CS) precedes and co-terminates with terminal photostimulation as an unconditioned stimulus (US) to assess the ability of activating each individual projection target to generate a fear memory, revealed by testing for conditioned responses to the CS in a novel environment (*Figure 5A*). Photostimulation of CGRP$^{PBN}$-neuron terminals in the VPMpc, PSTN, or SI resulted in significant freezing to the auditory CS after 6 CS-US pairings (*Figure 5B–G*), with only activation of terminals in the VPMpc or SI generating a significant association as indicated by area under the curve exceeding that of control animals (*Figure 5H*) and robust conditioned freezing to the CS in a novel context 24 hr following conditioning (*Figure 5B–G*). While photostimulation of CGRP$^{PBN}$ neuron terminals in either the SI or VPMpc was sufficient to drive associative fear learning, the association formed is weaker than that driven by photostimulating CGRP$^{PBN}$ neuron cell bodies (*Figure 5H*), suggesting they play complementary roles.

## Emergent properties of combined activation in multiple downstream targets

Activation of no single projection from CGRP$^{PBN}$ neurons was sufficient to elicit profound freezing behavior or bradycardia; therefore, we devised a method to simultaneously activate multiple terminal fields by implanting three fiber-optic cannulae in a single hemisphere over multiple areas of interest to determine the threshold of downstream activity necessary to elicit these phenotypes (*Figure 6A*). We placed one cannula over the SI, one over the cCeA and one over the VPMpc. Then, we determined the strength of freezing responses capable of being generated by each individual projection field by varying the light power. Maintaining stimulation frequency at 30 Hz and increasing laser power from 10 to 40 mW, we found that activation of CGRP$^{PBN}$ neuron terminals in the cCeA or VPMpc led to a gradual increase in freezing but activating terminals in the SI was maximal at 10 mW (*Figure 6B*). Combining photostimulation of terminals in the VPMpc and SI (10 mW each) led to rapid entrainment of freezing behavior to an auditory CS, and the resulting association strength, although not significantly greater than either projection individually generated, was no longer significantly weaker than that generated by the entire population even though our dual-stimulation arrangement was unilateral and all other groups were bilateral (*Figure 6C*).

We then combined activation of multiple projection fields using 20 mW power to determine which combination of CGRP$^{PBN}$ neuron projections could elicit profound freezing behavior. Activating the cCeA and SI projection fields resulted in moderate freezing behavior that did not appear to

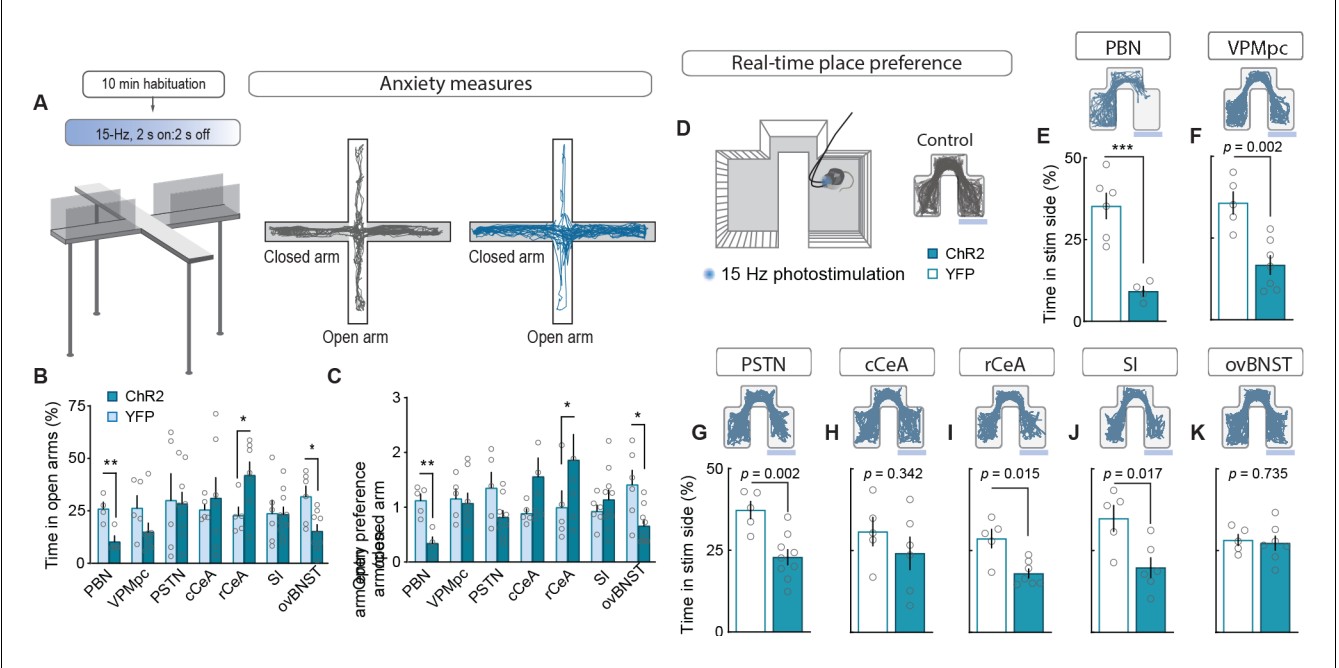

**Figure 4.** Stimulating CGRP[PBN] neuron terminals in ovBNST is anxiogenic while stimulating most other projections is aversive. (**A**) Experimental timeline and example responses to stimulation of CGRP[PBN] neuron terminals or somata during measurements of anxiety-like behavior. (**B**) Activation of CGRP[PBN] neurons reduced time spent in open arms, as did stimulation of terminals in the ovBNST. Activation of terminals in the rCeA increased open-arm exploration time. Significance for Welch's unpaired t-test (PBN $t(5.93)$ = 3.77, **p=0.009, n = 4.4 (ChR2, YFP); rCeA $t(9.42)$ = 2.59, *p=0.028, n = 7.5; ovBNST $t(8.85)$ = 2.65, *p=0.034, n = 9.6). (**C**) Activation of CGRP[PBN] neurons or their projection to the ovBNST reduced open-arm entry preference; activation of the projection to the rCeA increased open-arm entries. Significance for Welch's unpaired t-test (PBN $t(6.90)$ = 4.87, **p=0.002, n = 4.4; rCeA $t(5.59)$ = 2.51, *p=0.049, n = 7.5; ovBNST $t(6.87)$ = 2.89, *p=0.018, n = 9.6). (**D**) Illustration of RTPP paradigm and example trace of control mouse maze exploration. (**E**) Activation of CGRP[PBN] neurons led to avoidance of light-paired side (Welch's unpaired t-test, $t(6.31)$ = 6.27, ***p<0.001, n = 6.4). (**F**) Mice avoid photostimulation of CGRP[PBN] neuron terminals in the VPMpc (Welch's unpaired t-test, $t(8.75)$ = 4.28, p=0.002, n = 7.5). (**G**) Mice avoid photostimulation of CGRP[PBN] neuron terminals in the PSTN (Welch's unpaired t-test, $t(9.71)$ = 4.11, p=0.002, n = 9.5). (**H**) Photostimulation of CGRP[PBN] neuron terminals in the cCeA does not affect place-preference (Welch's unpaired t-test, $t(8.99)$ = 1.00, p>0.05, n = 6.5). (**I**) Mice avoid photostimulation of CGRP[PBN] neuron terminals in the rCeA (Welch's unpaired t-test, $t(5.92)$ = 3.38, p=0.015, n = 7.5). (**J**) Mice avoid photostimulation of CGRP[PBN] neuron terminals in the SI (Welch's unpaired t-test, $t(7.87)$ = 3.02, p=0.017, n = 6.5). (**K**) Photostimulation of CGRP[PBN] neuron terminals in the ovBNST does not affect place-preference (Welch's unpaired t-test, $t(9.99)$ = 0.35, p>0.05, n = 7.5). Data are represented as mean ± SEM. For full statistical information see *Supplementary file 1*.

The online version of this article includes the following figure supplement(s) for figure 4:

**Figure supplement 1.** Activation of CGRP[PBN] terminals in the rCeA potentiates nocifensive responses.

be additive (cCeA 23.2 ± 0.5% freezing; SI 26.9 ± 2.5; Combined 30.4 ± 1.3 (mean ± sem), *Figure 6D*; *Figure 6—video 1*), while simultaneous activation of terminals in the VPMpc and cCeA elicited robust freezing behavior exceeding that produced individually (VPMpc 37.1 ± 2.3% freezing; cCeA 23.2 ± 0.5; Combined 68.9 ± 3.6 (mean ± sem), *Figure 6B and E–F*; *Figure 6—video 2*), comparable to freezing behavior elicited by activating all CGRP[PBN] neurons bilaterally (93.0 ± 2.9% freezing (mean ± sem), *Figure 1B*; *Figure 6—video 1*). These phenotypes were enhanced by driving photostimulation with a red light-activated opsin (*Yizhar et al., 2011*) (VPMpc+cCeA 94.5 ± 3.0% freezing (mean ± sem), *Figure 6—figure supplement 1A–I*), suggesting that a combination of light-spread and faithfulness of activation underlies reliable freezing generation. Importantly, simultaneous photostimulation of terminal fields did not dramatically induce Fos in CGRP[PBN] neurons (<10% compared to 80% for soma activation sufficient to generate freezing behavior) (*Figure 6—figure supplement 1J–L*). We also tested whether simultaneous photostimulation of terminals in the cCeA and VPMpc would affect autonomic physiology by measuring heart rate using a pulse oximeter. While activating neither projection alone affected heart rate (*Figure 3*), simultaneous photostimulation robustly elicited bradycardia, which consistently took longer to develop than freezing responses (*Figure 6G*, *Figure 6—figure supplement 1A*). These results imply that when

combinations of projections from CGRP<sup>PBN</sup> neurons are activated simultaneously their combined output is able to generate phenotypes beyond their individual ability, suggesting a mechanism by which defensive responses can be tuned depending on whether downstream partners are already in an 'up' or 'down' state as determined by broader sensory context when they receive input from CGRP<sup>PBN</sup> neurons.

## Foot shock-induced activation of the VPMpc and SI by CGRP<sup>PBN</sup> neurons contributes to associative fear learning

While previous studies that permanently silenced CGRP<sup>PBN</sup> neurons demonstrated that their activity contributes to conditioned-fear responses (*Han et al., 2015*), we asked whether photoinhibition restricted to the peri-foot shock period during conditioning would be sufficient to attenuate conditioned responses to the CS, as post-shock recurrent activity, stress-induced activation, or recall-driven reactivation could also potentially affect association formation, memory consolidation, or recall. Using AAV-mediated expression of a red-light activated chloride pump (*Chuong et al., 2014*) (JAWS) to inhibit CGRP<sup>PBN</sup> neurons during 0.5-mA foot-shock delivery (*Figure 7A*, *Figure 7—figure supplement 1A*), we found that selective inhibition of CGRP<sup>PBN</sup> neurons during the foot shock significantly attenuated both conditioned responses during training and in a CS-probe trial 24 hr later, while also reducing freezing behavior conditioned to the training context (*Figure 7B*). These findings affirm that the signal relayed by CGRP<sup>PBN</sup> neurons to downstream partners during the foot shock directly contributes to associative memory formation.

To determine whether individual projections contribute to associative fear learning, we used JAWS to inhibit CGRP<sup>PBN</sup>-neuron terminals in the VPMpc, CeA, or SI during the foot shock (*Figure 7C*). We first confirmed that JAWS-mediated inhibition of CGRP<sup>PBN</sup>-neuron terminals significantly reduced EPSC frequency in post-synaptic neurons (*Figure 7D*; *Mahn et al., 2016*). Inhibiting synaptic release during the foot shock at CGRP<sup>PBN</sup>-neuron terminals in the VPMpc or SI, but not CeA, significantly attenuated both memory formation (*Figure 7E–G*) and association strength (*Figure 7H*), without affecting contextual-fear learning. While inhibiting CGRP<sup>PBN</sup> neurons non-significantly reduced foot shock-induced locomotion (*Figure 7I*), no individual projection tested was necessary for this response. In addition, transiently inhibiting either CGRP<sup>PBN</sup> cell bodies or their individual projections did not significantly affect behavioral responses to noxious heat (*Espejo and Mir, 1993*; *Figure 7—figure supplement 1B–E*), nor did it lead to a place preference in a RTPP paradigm (*Figure 7—figure supplement 1F–I*), suggesting that basal activity of CGRP<sup>PBN</sup> neurons is insufficient for their inhibition to generate a salient shift in affective state. Taken together, these data reveal an unexpected role for the SI and VPMpc, two regions respectively implicated in arousal (*Kaur et al., 2017*; *Mogenson et al., 1985*) and taste processing (*Liu and Fontanini, 2015*), in contributing to an affective pain signal that drives associative fear learning.

## Discussion

Disentangling the interacting neural substrates responsible for generating affective, behavioral, and physiological responses to environmental threats is a necessary endeavor for understanding and eventually treating the alterations in threat processing that underlie affective disorders such as PTSD (*Flandreau and Toth, 2018*; *Mikics et al., 2008*) and anxiety (*Davis and Whalen, 2001*; *Lissek et al., 2014*). Leveraging what is known about the circuits ascending from the spinal cord to drive affective, motivational responses to pain (*Bernard and Besson, 1988*; *Campos et al., 2018*; *Gauriau and Bernard, 2002*; *Han et al., 2015*), we aimed to dissect at the level of the PBN the multi-faceted system that simultaneously generates diverse innate unconditioned responses and drives learned associations to aversive stimuli.

### Generation of unconditioned behavioral and physiological responses

Previous studies silencing CGRP<sup>PBN</sup> neurons implicated them in contributing to both affective responses to somatic pain, including nocifensive behavior, post-shock freezing behavior (*Han et al., 2015*), and illness-induced increases in anxiety state (*Campos et al., 2017*). We found that photostimulation of CGRP<sup>PBN</sup> neurons, in addition to driving profound freezing behavior, can also generate either tachycardia or parasympathetic responses depending on stimulation frequency, and elicit anxiety-like behavior. These findings collectively suggest that activation of CGRP<sup>PBN</sup> neurons during

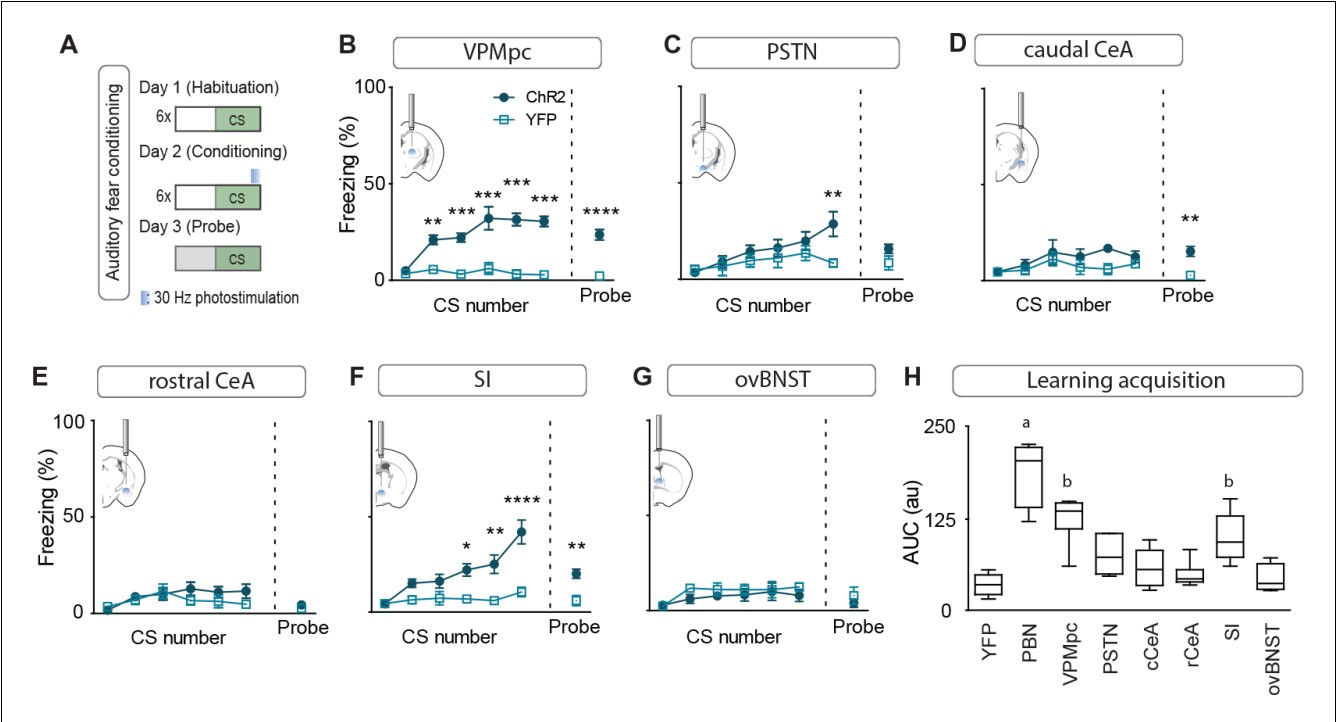

**Figure 5.** Photostimulating terminals in the VPMpc or SI can promote associative fear learning. (A) Illustration of experimental paradigm for cue-dependent optogenetic conditioning. (B) Conditioned-freezing responses to CS paired with CGRP[PBN] terminal stimulation in the VPMpc during training (n = 8.5; ChR2, YFP; significant effect of group in two-way ANOVA, $F_{1,66}$ = 115.4, p<0.0001; subsequent Sidak pairwise comparisons, **p<0.01; ***p<0.001) and in probe test 24 hr following conditioning (Welch's unpaired t-test, t(8.93) = 7.29, ****p<0.0001). (C) Conditioned freezing responses to PSTN (n = 5.4) terminal stimulation (significant effect of group in two-way ANOVA, $F_{1,42}$ = 6.99, p=0.012; subsequent Sidak pairwise comparisons). (D) Conditioned freezing responses to cCeA (n = 7.5) terminal stimulation (significant group effect in two-way ANOVA during training, $F_{1,60}$ = 4.69, p=0.0343; and probe test, Welch's unpaired t-test, t(8.15) = 4.40, **p=0.0022). (E) Conditioned freezing responses to rCeA (n = 8.5) terminal stimulation (two-way ANOVA effect of group, $F_{1,60}$ = 2.74, p=0.1032). (F) Conditioned freezing responses to SI (n = 8.6) terminal stimulation. Significant group effect in two-way ANOVA during training, $F_{1,60}$ = 23.45, p=0.0004; subsequent Sidak pairwise comparisons; and in probe test 24 hr following conditioning (Welch's unpaired t-test, t(11.15) = 3.86, **p=0.0026). (G) Conditioned freezing responses to ovBNST (n = 5.4) terminal stimulation (two-way ANOVA effect of group, $F_{1,66}$ = 2.764, p=0.1011). (H) Area under the curve for conditioning in each ChR2 fiber-placement group, including PBN-stimulation (n = 8) and control groups (n = 6, averaged for each YFP fiber-placement group). Significance for one-way ANOVA, $F_{7,40}$ = 19.44, p<0.0001; subsequent Tukey correction for multiple comparisons, differences indicated by dissimilar letters above data columns. Bar graphs are represented as mean ± SEM. For full statistical information see *Supplementary file 1*.

somatic pain has the potential to contribute to many aspects of the unconditioned response cascade, from shock-induced locomotion to post-shock freezing behavior, autonomic responses including simultaneous enhancement of parasympathetic and sympathetic outflow, and post-insult anxiogenesis. A complication of this arrangement is that neither freezing behavior (*Blanchard and Blanchard, 1969*), parasympathetic responses (*Iwata and LeDoux, 1988*), nor anxiety occur during the shock. Hence, the role played by CGRP[PBN] neurons in these phenotypes would necessarily result from recurrent reactivation, rather than a direct ascending signal.

By selectively activating CGRP[PBN]-neuron terminals in their various downstream targets, we distinguished the potential of individual downstream partners to contribute to distinct components of the behavioral and physiological alterations that comprise the unconditioned- response cascade. We found that with the exception of the rCeA, all projections generated some amount of freezing behavior, with the most robust responses elicited by the PSTN and VPMpc, two projections that were overlooked in previous work. We also found a marked disparity in function across the CeA, with activation of terminals in the cCeA eliciting only mild freezing behavior, while activating the rCeA had no effect on freezing behavior during photostimulation but did produce robust sympathetic responses, brief contextual freezing following stimulation offset, avoidance, and nocifensive behaviors on a hot plate, all phenotypes reminiscent of responses to noxious stimulation. In general,

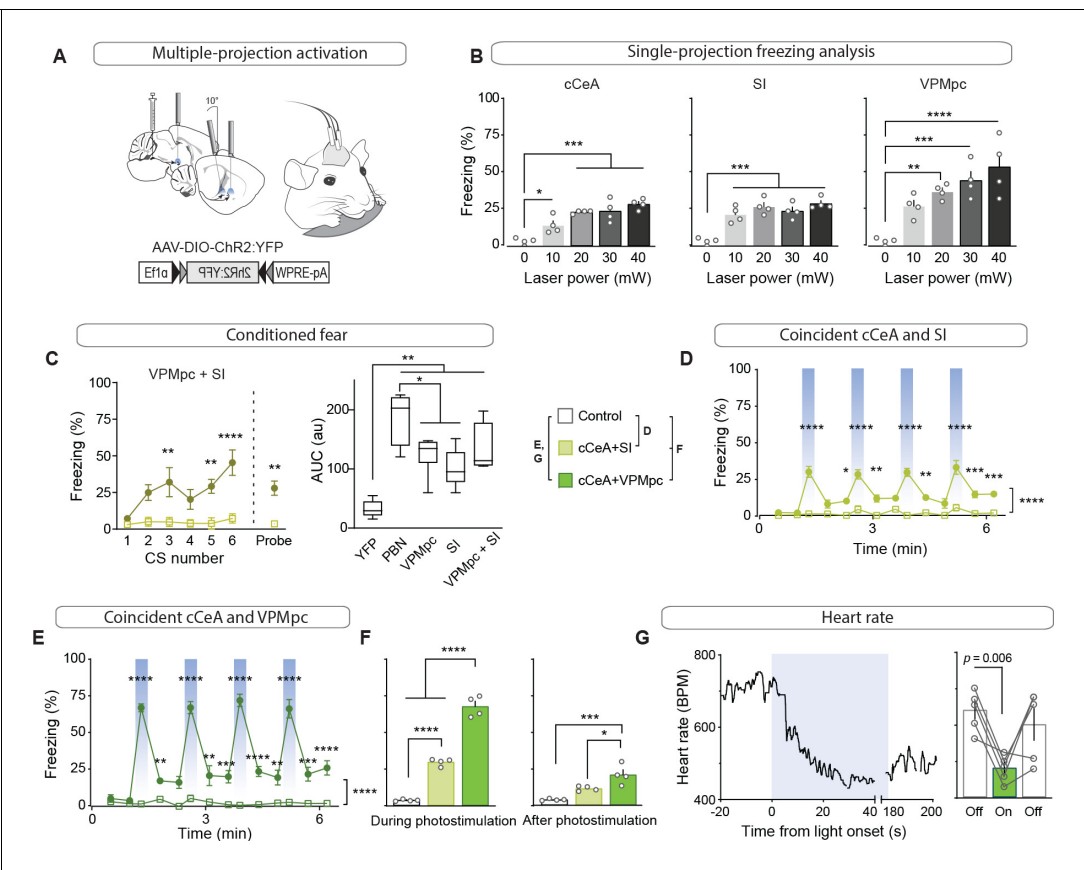

**Figure 6.** Combined activation of CGRP[PBN] neuron terminals in the VPMpc and cCeA scales freezing responses and produces bradycardia. (**A**) Schematic showing configuration for implantation of 3 fiberoptic cannulae into one hemisphere allowing simultaneous photostimulation of multiple CGRP[PBN]-neuron terminal fields. (**B**) Freezing behavior during 30 Hz photostimulation with increasing power of CGRP[PBN] neuron terminal fields in the cCeA (n = 4, one-way ANOVA, $F_{4,15}$ = 18.08, p<0.0001), SI (n = 4, $F_{4,15}$ = 19.21, p<0.0001), or VPMpc (n = 4, $F_{4,15}$ = 12.09, p=0.0001). Subsequent Tukey correction for multiple comparisons, *p<0.05; **p<0.01; ***p<0.01; ****p<0.0001. (**C**) Freezing behavior to auditory CS co-terminating with simultaneous photostimulation of terminals in the VPMpc and SI (left) (significant group effect in two-way ANOVA, $F_{1,6}$ = 21.57, p=0.0035; subsequent Sidak pairwise comparisons) or probe test with CS presented in novel context 24 hr after conditioning (Welch's unpaired t-test, $t(3.414)$ = 4.90, p=0.012). Comparison of area-under-curve for associative learning generated by CS paired with CGRP[PBN] neuron or terminal activation (right) (one-way ANOVA, $F_{4,27}$ = 19.73, p<0.0001); subsequent Tukey correction for multiple comparisons. Center line, mean; box limits, upper and lower quartiles; whiskers, min to max. (**D**) Freezing behavior in response to simultaneous activation of CGRP[PBN] neuron terminals in the cCeA and SI (n = 4.4 (ChR2, control), significant group effect in two-way ANOVA, $F_{1,78}$ = 213.5, p<0.0001; subsequent Sidak pairwise comparisons). (**E**) Freezing behavior in response to simultaneous activation of CGRP[PBN] neuron terminals in the caudal CeA and VPMpc (n = 4.4; significant group effect in two-way ANOVA, $F_{1,84}$ = 631.5, p<0.0001; subsequent Sidak pairwise comparisons). (**F**) Comparison of averaged freezing behavior for each stimulation combination during the stimulation epoch (left) (n = 4.4; one-way ANOVA, $F_{2,9}$ = 218.9, p<0.0001), and during the post-stimulation epoch (right) (n = 4,4; one-way ANOVA, $F_{2,9}$ = 17,67, p=0.0008; subsequent Tukey correction for multiple comparisons. (**G**) Representative (left) and mean bradycardia elicited by simultaneous photostimulation of CGRP[PBN]-neuron terminals in the cCeA and VPMpc (n = 5; one-way ANOVA, $F_{2,12}$ = 7.38, p=0.0081; subsequent Dunnett correction for multiple comparisons, p=0.0058). Bar graphs represented as mean ± SEM. See also *Figure 6—videos 1* and *2*. For full statistical information see *Supplementary file 1*.

The online version of this article includes the following video and figure supplement(s) for figure 6:

**Figure supplement 1.** Coincident activation of CGRP[PBN] neuron projections using ChrimsonR causes profound freezing responses.

**Figure 6—video 1.** Freezing behavior generated by activating the SI and caudal CeA simultaneously supplement to *Figure 6*.
https://elifesciences.org/articles/59799#fig6video1

**Figure 6—video 2.** Freezing behavior generated by activating the caudal CeA and VPMpc simultaneously supplement to *Figure 6*.
https://elifesciences.org/articles/59799#fig6video2

our results suggest that CGRP[PBN]-neuron connections to extended amygdalar structures (i.e. the CeA, SI, and ovBNST) influence freezing behavior, affective processing including negative valence and anxiety state, and physiological responses, while thalamic and hypothalamic connections

transmit a negative-valence signal and elicit freezing behavior. These results are supported by the fact that, in rats, extended amygdalar structures are richly interconnected with hindbrain nuclei controlling autonomic outflow (*Dong and Swanson, 2004*; *Rizvi et al., 1991*; *Veening et al., 1984*), while the VPMpc is not (*Cechetto and Saper, 1987*). Our findings complement recent work that distinguished between PBN populations that target the extended amygdala and hypothalamus/periaqueductal grey and differentially drive affective and nocifensive responses, respectively (*Chiang et al., 2020*). However, by distinguishing between conditioned freezing responses and avoidance behavior, we were able to specifically implicate two novel targets, the VPMpc and SI, in generating associative fear learning, whereas *Chiang et al., 2020* studies were limited to learned valence and only tested the CeA and ovBNST. We found that the PSTN, SI, and VPMpc also relay a negative valence signal. Hence, populations contributing to the negative valence of noxious or aversive stimuli may not necessarily contribute to associative fear learning, which has important implications for understanding the neural underpinnings of affective disorders such as PTSD.

Our collateral-tracing experiments revealed that, in contrast to the distinct phenotypes generated by terminal photostimulation, CGRP$^{PBN}$ neurons form a broadly distributed network with their downstream partners in which no forebrain target receives solitary innervation. There was some bias in the connectivity groupings, with neurons projecting to the CeA tending to also strongly innervate the PSTN, neurons projecting to the VPMpc also innervating the SI and IC and avoiding the ovBNST, and neurons projecting to the ovBNST also targeting the CeA. Our findings are broadly in agreement with previous experiments that delineated sub-region-specific output and collateralization in rats (Sarhan et al.) and mice (*Chiang et al., 2020*), neither of which, however, reported connections or collateralization with the VPMpc, suggesting that cell-type specific expression more efficiently reveals this connection. Of interest, *Sarhan et al., 2005* beautifully outlined rostral and caudal capsular CeA branching patterns across all extended amygdalar structures using single-axonal reconstructions of PBN neurons, finding collateralization between the rCeA and lateral hypothalamus (PSTN), and cCeA and ventral BNST. Taken as a whole, the distributed, collateralization organization of CGRP$^{PBN}$ neurons may be important for generating highly coordinated actions and associations by simultaneously driving activity in downstream sites that have related or complementary functions. An example in support of this arrangement is that stimulation of terminals in the SI and VPMpc generated disparate effects on physiology, but collaboratively supported associative fear learning.

While activating some individual terminal fields from CGRP$^{PBN}$ neurons in different downstream sites recapitulated – in a scaled-down fashion – most of the phenotypes driven by photostimulating the cell bodies, we found that profound freezing behavior and bradycardia were not produced by stimulation of any individual projection, suggesting they instead arise from additive interactions between downstream structures and their respective circuits. We tested this hypothesis by simultaneously activating terminals in the VPMpc and cCeA, two targets that generated reliable freezing behavior, and observed not only a robust potentiation of the freezing behavior but also profound bradycardia. Interestingly, neither of these populations generated autonomic responses when activated individually. One possible arrangement that explains this phenotype is that their concurrent activation gates activity in secondary structures that drive parasympathetic responses.

An important consideration in implicating individual downstream partners in generating distinct aspects of behavioral and physiological response is the inherent limitation of terminal photostimulation. It is difficult, if not impossible, to ensure that antidromic activity does not activate secondary targets, an especially important possibility given the broad collateralization of CGRP$^{PBN}$ neurons. However, secondary techniques aimed at accounting for this situation also have their shortcomings: axons may bifurcate near the sites of interest rather than at the cell body, hence silencing cell bodies may not prevent antidromic activation. Moreover, since many of the forebrain structures contributing to threat processing are interconnected, silencing other portions of the downstream circuit to attempt to isolate the effect of the target of interest on the measured phenotype may affect phenotype generation if the populations are interconnected. We argue that the very fact that terminal stimulation in different downstream targets generates distinct phenotypes supports the fact that at minimum, preferential activation of the site of interest is occurring. If photostimulation of terminals was efficiently activating cell bodies within the PBN then the same phenotypes should be observed regardless of fiber location. Perhaps, most compelling is that CeA-projecting CGRP$^{PBN}$ neurons make up the bulk of the population yet photostimulation of terminals in the CeA does not efficiently

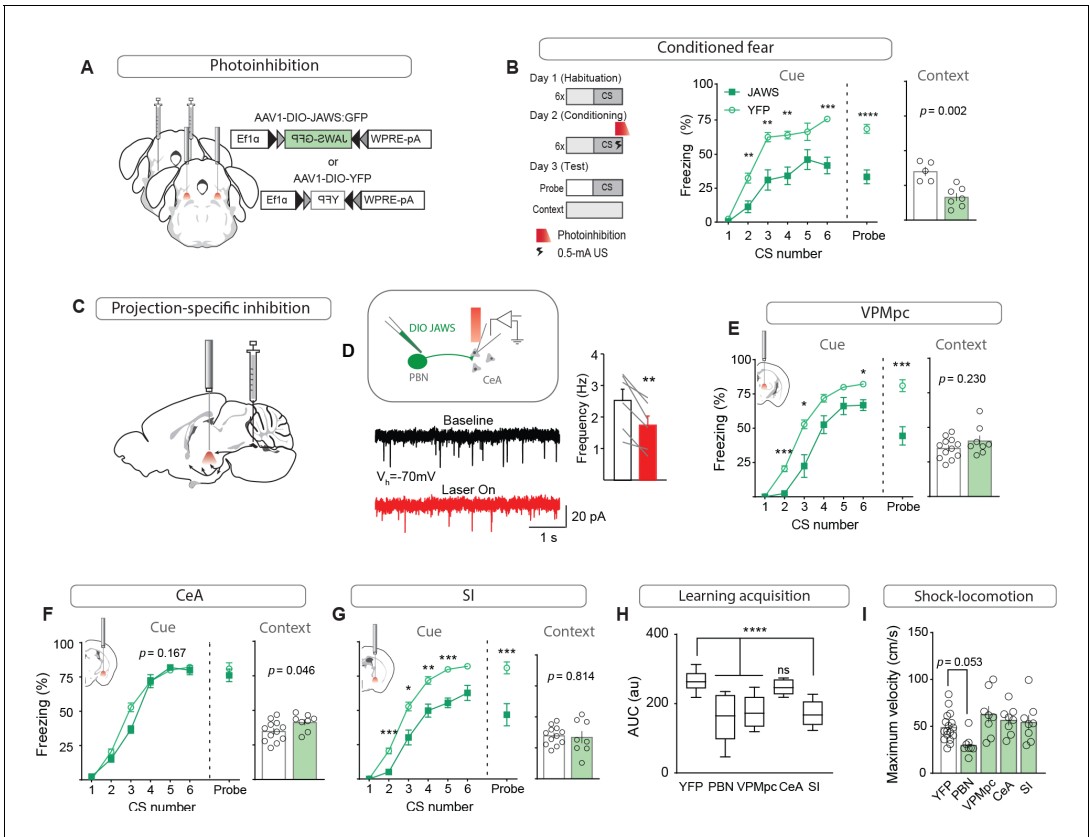

**Figure 7.** Foot shock-induced activation of the VPMpc and SI by CGRP[PBN] neurons contributes to associative fear learning. (A) Bilateral injections of AAV1-DIO-JAWS:GFP or AAV1-DIO-YFP and fiber-optic cannula implants above the PBN of Calca[Cre/+] mice for photoinhibition of CGRP[PBN] neurons. (B) Photoinhibition of CGRP[PBN] neurons (n = 8,5; JAWS, GFP) during foot shock delivery attenuated freezing responses both to CS and context (significant group effect in two-way ANOVA for training, $F_{1,55}$ = 21.66, p=0.0007; subsequent Sidak pairwise comparisons, *p<0.05; **p<0.01; Welch's unpaired t-test for probe and context, probe $t(10.88)$ = 6.45, ****p<0.0001; context $t(8.93)$ = 4.43, p=0.002). (C) Placement of fiber-optics over projection sites for projection-specific photoinhibition. (D) Representative recording of EPSCs in a CeA neuron surrounded by JAWS:GFP-positive fibers from CGRP[PBN] neurons. Red light decreased frequency of EPSCs in downstream cells (6 cells from two mice, paired t-test, $t(5)$ = 4.84, **p=0.0047). (E) Photoinhibition of CGRP[PBN] neuron terminals in the VPMpc (n = 8,12) during footshock attenuated freezing responses to CS (significant group effect in two-way ANOVA for training, $F_{1,18}$ = 28.78, p<0.0001; subsequent Sidak pairwise comparisons; probe test Welch's unpaired t-test, $t(14.41)$ = 4.58, ***p=0.0004) but not context (Welch's unpaired t-test, $t(11.72)$ = 1.27, p>0.05). (F) Effect of photoinhibition of CGRP[PBN] neuron terminals in the CeA (n = 8,12) during foot shock on conditioned freezing responses to cue (two-way ANOVA for training, group effect $F_{1,18}$ = 2.08, p=0.167; Welch's unpaired t-test for probe, $t(16.69)$ = 0.76, p=0.46) or context (Welch's unpaired t-test, $t(17.17)$ = 2.15, p=0.046). (G) Photoinhibition of CGRP[PBN] neuron terminals in the SI (n = 8.12) during shock attenuated freezing responses to CS (significant group effect in two-way ANOVA for training, $F_{1,18}$ = 40.52, p<0.0001; subsequent Sidak pairwise comparisons; probe test Welch's unpaired t-test, $t(11.06)$ = 3.70, **p=0.0035) but not the context (Welch's unpaired t-test, $t(9.83)$ = 0.24, p>0.05). (H) Photoinhibition of CGRP[PBN] neuron projections to either the VPMpc or SI during footshock attenuated associative learning (area under learning curve) as efficiently as silencing the entire population. Center line, mean; box limits, upper and lower quartiles; whiskers, min to max. Significance for one-way ANOVA, $F_{4,45}$ = 15.35, p<0.0001; subsequent Sidak pairwise comparisons found no difference between PBN, VPMpc, and SI fiber-placement groups. (I) Locomotion during foot shock was not significantly affected by photoinhibition of CGRP[PBN] neurons (one-way ANOVA, $F_{4,44}$ = 4.13, p=0.0063; subsequent Dunnett pairwise comparisons p>0.05). Bar graphs are represented as mean ± SEM. For full statistical information see ***Supplementary file 1***.

The online version of this article includes the following figure supplement(s) for figure 7:

**Figure supplement 1.** Photoinhibition of CGRP[PBN] neurons or projections does not affect nocifensive responses or alter place preference.

produce either freezing behavior or anxiety, two of the distinct phenotypes produced by activating other downstream targets that receive collateral innervation with the CeA.

Another necessary caveat of using artificial stimulation to probe the intrinsic functionality of different projection partners is that variation in transport rate and axon length across downstream partners could profoundly influence ChR2 levels at terminals and thus terminal stimulation efficacy, which would in turn confound the observed differences in function and connection strength across

the circuit. We attempted to account for this by allowing 4 weeks after virus injection for ChR2 expression and transport before beginning experiments, having observed that 2–3 weeks was sufficient to observe robust labeling in the most distant terminal fields. While our electrophysiological measures revealed the greatest synaptic strength in CGRP$^{PBN}$ neuron connections to one of their more proximal downstream partners, the VPMpc, the fact that the PSTN, which is equally close to the PBN but has a connection strength that is similar to what we observed with all other targets suggests that ChR2 transport cannot be the primary reason for the strength of the connection to the VPMpc. Moreover, we did not see a declination of synaptic strength across the proximal-distal axis – synapses in the ovBNST, the most distal target, exhibited similar synaptic strength to those 2 mm more proximal, in the PSTN. What then underlies the large differences in synaptic strength we observed across downstream targets that was independent of projection strength as measured by terminal labeling density? Differences in release probability, convergence (the VPMpc is a much smaller structure than the CeA, for example, hence each neuron might receive more contacts), post-synaptic receptor number or dendrite structure could all contribute to the observed differences in synaptic strength. One particularly interesting possibility is that the VPMpc, which does not express receptors for CGRP, may primarily rely on glutamatergic input from the PBN and hence exhibit large-amplitude EPSCs in response to CGRP$^{PBN}$ neuron terminal photostimulation, while extended amygdala targets may rely more on CGRP release for activation (*Shinohara et al., 2017*; *Okutsu et al., 2017*), which our experiment would not have been able to reveal. A more detailed analysis of the electrophysiological properties of CGRP$^{PBN}$ neuron to forebrain connections is necessary to understand the temporal and chemical underpinnings that help give rise to the different functions of each downstream connection.

## Associative fear learning

Associative learning is a highly tractable and informative process because it reliably depends on the salience of the CS and US, and the innate associability of these stimuli (*Garcia et al., 1968*; *Sigmundi et al., 1980*). The interplay of these factors on the association is indicated by the learning rate and asymptote – the maximal conditioned response for a particular CS-US pair (*Rescorla, 1972*; *Sigmundi et al., 1980*). Here, we maintained a constant CS and varied the US by activating specific projections from CGRP$^{PBN}$ neurons to condition predictive freezing to an auditory CS, or by silencing either CGRP$^{PBN}$ neurons or individual projections during foot-shock delivery. Activation of CGRP$^{PBN}$ neurons elicited the most robust association, followed by stimulation of terminals in the VPMpc or SI. No individual projection was sufficient to recapitulate the learning asymptote generated by stimulating all CGRP$^{PBN}$ neurons; hence, some combination of projections relays salient aspects of the US, generating complementary signals that eventually reach the basolateral amygdala (BLA) to potentiate synapses receiving coincident CS information (*Blair et al., 2001*; *Maren, 2001*; *Romanski et al., 1993*). In support of this hypothesis, inhibiting CGRP$^{PBN}$-neuron terminals in either the VPMpc or SI during the US attenuated the association strength to the same degree as inhibiting the entire population, suggesting that preventing activation of either downstream partner impairs associative learning. We also observed that a substantial degree (~50%) of the conditioned response was maintained when inhibiting CGRP$^{PBN}$ neurons, indicating that they are part of a distributed network that collectively relays the affective, motivational signal to forebrain neurons that form and store the associative memory (*Lanuza et al., 2004*; *Lanuza et al., 2008*; *Shi and Davis, 1999*). Interestingly, work examining the ability of the CGRP$^{PBN}$ neuron projection to the VPMpc to generate an associative memory using taste as a CS indicated no conditioned taste aversion formation when paired with brief photostimulation, while projections to the CeA and ovBNST did (*Chen et al., 2018*), suggesting that either the relayed signal is the wrong modality for combining with CS taste information, or that a specific temporal activation pattern different from that tested is required to form an association (e.g. longer term activation that better mimics visceral illness). Hence, it is surprising that associative learning to a tone generated conditioned freezing behavior, while association with a taste did not alter preference, even though the VPMpc is an integral part of the ascending taste network (*Liu and Fontanini, 2015*). More work assessing the response patterns of individual VPMpc neurons to diverse sensory modalities and their contribution to conditioned taste aversion is required to resolve these paradoxes.

Previous work in rats indicates that PBN projections to the CeA are capable of eliciting both escape behaviors and fear learning in addition to driving place avoidance (*Sato et al., 2015*), while

in mice it has been demonstrated that activation of CGRP-receptor neurons in the CeA is sufficient to act as a US to drive associative fear learning, and that silencing these neurons prior to conditioning attenuates conditioned-fear responses (*Han et al., 2015*). We activated both the rostral and caudal CeA terminal fields of CGRP[PBN] neurons and were surprised that neither manipulation was individually capable of generating a fear memory, suggesting a number of possibilities that may give rise to these findings: (1) It is possible that CGRP[PBN] neurons are only one component of the PBN projection to CeA and other PBN cell types are important for aversive learning, (2) CGRP-receptor neurons in the CeA may make up a larger population than those activated by CGRP[PBN]-neuron terminal stimulation and receive input from secondary sources that are important for aversive learning; (3) direct activation of neurons in the CeA may be more efficient than terminal stimulation and may thus be able to drive associative learning; and (4) since the mice used in our study are heterozygous for CGRP, wholly intact neuropeptide signaling may be required for association formation derived from PBN to CeA stimulation (*Shinohara et al., 2017*; *Okutsu et al., 2017*). In support of the first two possibilities, we observed that inhibiting CGRP[PBN] neuron terminals in the CeA during foot-shock delivery had no effect on associative fear learning, suggesting that the relayed activity from CGRP[PBN] neurons to the CeA during the foot shock is not necessary for the CS-US association. This is in apparent contrast to previous work demonstrating that silencing CGRP-receptor neurons in the CeA prior to conditioning attenuates conditioned responses to the CS (*Han et al., 2015*). However, that manipulation was permanent and failed to distinguish between association formation and recall, which our US-only inhibition did, suggesting that reactivation of CGRP-receptor neurons in the CeA after conditioning may underlie the observed reductions in conditioned responding. Based on these observations and our data implicating CGRP[PBN] neuron projections to the CeA in generating robust unconditioned-responses, we propose an alternate model, wherein CGRP[PBN] neuron connections to the CeA, PSTN, SI, and ovBNST drive unconditioned responses to the US, post-conditioning activation of BLA neurons by the CS reactivates the CeA to generate conditioned responses (*Kim et al., 2017*), and CGRP[PBN] neuron connections to the SI and VPMpc primarily mediate the role of CGRP[PBN] neurons in associative-fear learning, a compelling arrangement given that in rats, these two downstream partners are the most directly invested in cortical circuits (*Cechetto and Saper, 1987*; *Wenk, 1997*). Our data establish partially separable ascending routes from CGRP[PBN] neurons for generating unconditioned responses and forming associative memories to aversive stimuli.

## Materials and methods

### Key resources table

| Reagent type (species) or resource | Designation | Source or reference | Identifiers | Additional information |
|---|---|---|---|---|
| Strain, strain background (*Mus musculus*) | Calca-Cre, C57BL6/J | *Carter et al., 2013* | RRID:IMSR_JAX:033168 | |
| Strain, strain background (*AAV1*) | pAAV1-Ef1alpha-DIO hChR2(H134R) eYFP | *Carter et al., 2013* | Addgene Plasmid #20298 RRID:Addgene_20298 | |
| Strain, strain background (*AAV1*) | pAAV1-Ef1alpha-DIO YFP | *Carter et al., 2013* | Addgene Plasmid #27056 RRID:Addgene_27056 | |
| Strain, strain background (*AAV1*) | pAAV1-Ef1alpha-DIO JAWS GFP | *Jo et al., 2018* | RRID:Addgene_78174 | |
| Strain, strain background (*AAV1*) | pAAV1-nEF-Con/Fon-ChR2-mCherry | *Fenno et al., 2014* | Addgene Plasmid #137142 RRID:Addgene_137142 | |
| Strain, strain background (*AAV1*) | pAAV1-nEF-Con/Foff 2.0-ChR2-mCherry | *Fenno et al., 2014* | Addgene Plasmid #137143 RRID:Addgene_137143 | |
| Strain, strain background (*rAAV2-retro*) | AAV2-retro-CBA-Flippase-dsRed | This paper | N/A | palmiter@uw.edu |
| Strain, strain background (*AAV9*) | AAV9-Syn-ChrimsonR-tdTomato | UNC Vector Core | Cat# AV6556B A RRID:Addgene_62723 | |

*Continued on next page*

Continued

| Reagent type (species) or resource | Designation | Source or reference | Identifiers | Additional information |
|---|---|---|---|---|
| Antibody | Anti-c-Fos (Rabbit polyclonal) | Abcam | Cat#: ab190289 RRID:AB_2737414 | (1:1000) |
| Antibody | Anti-GFP (Chicken polyclonal) | Abcam | Cat#: ab13970 RRID:AB_300798 | (1:10,000) |
| Antibody | Anti-dsRed (Rabbit monoclonal) | Takara | Cat#: 632496 RRID:AB_10013483 | (1:1000) |
| Antibody | Alexa Fluor 488 anti-Chicken (Donkey monoclonal) | Jackson ImmunoResearch | Cat#: 703-545-155 RRID:AB_2340375 | (1:500) |
| Antibody | Alexa Fluor 594 anti-Rabbit (Donkey monoclonal) | Jackson ImmunoResearch | Cat#: 711-585-152 RRID:AB_2340621 | (1:500) |
| Antibody | Cy5 anti-rabbit (Donkey monoclonal) | Jackson ImmunoResearch | Cat#: 711-175-152 RRID:AB_2340607 | (1:500) |
| Other | Normal donkey serum | Jackson ImmunoResearch | Cat#:017-000-121 RRID:AB_2337258 | |
| Other | Pulse oximeter | STARR Life Sciences | Part#: 015000 | |
| Other | Pulse oximeter collar sensor | STARR Life Sciences | Part#: 015021 | |
| Software, algorithm | MouseOxPlus conscious applications module | STARR Life Sciences | Part#: 015002 | |
| Software, algorithm | Ethovision XT 10 | Noldus Technology | www.noldus.com RRID:SCR_000441 | |

## Animals

*Calca*$^{Cre/+}$ mice (C57Bl/6 background) were generated and maintained as described (*Carter et al., 2013*). Male and female *Calca*$^{Cre/+}$ mice were used for all studies. Following stereotaxic surgery, mice were singly housed for at least three wk prior to and during experimentation with ad libitum access (unless noted otherwise) to standard chow diet (LabDiet 5053) in temperature- and humidity-controlled facilities with 12 hr light/dark cycles. All animal care and experimental procedures were approved by the Institutional Animal Care and Use Committee at the University of Washington.

## Virus production

AAV9-Flex-ChrimsonR:tdTomato was purchased from UNC GTC Vector Core (AV6556B; $4.5 \times 10^{12}$ viral particles/mL). AAV1-DIO-ChR2:YFP, AAV1-DIO-JAWS:GFP, rAAV2-retro-Flp, AAV1-Cre$_{on}$-Flp$_{off}$-ChR2-YFP, AAV1-Cre$_{on}$-Flp$_{on}$-ChR2-YFP and AAV1-DIO-YFP viral vectors were produced in-house by transfecting HEK cells with each of these plasmids plus pDG1 (AAV1 coat stereotype) helper plasmid; viruses were purified by sucrose and CsCl-gradient centrifugation steps, and re-suspended in 0.1 M phosphate-buffered saline (PBS) at about $10^{13}$ viral particles/ml.

## Stereotaxic surgery

Bilateral stereotaxic injections of virus (0.28 µl per side) into the PBN of *Calca*$^{Cre/+}$ mice were achieved as described (*Carter et al., 2013*). In mice used for ChR2-optogenetic experiments, two custom-made fiber-optic cannulas were implanted bilaterally above the PBN (AP 4.70 mm, ML ±1.50 mm, DV 2.90 mm), VPMpc (AP 1.90 mm, ML ±1.25 mm, DV 3.65 mm), PSTN (AP −1.80 mm, ML ±1.50 mm, DV 4.60 mm), cCeA (AP 1.50 mm, ML ±3.10 mm, DV 4.30 mm), rCeA (AP 0.70 mm, ML ±2.85 mm, DV 4.50 mm), SI (AP 0.30 mm, ML ±1.80 mm, DV 4.40 mm), or BNST (AP +0.20 mm, ML ±1.20 mm, DV 4.00 mm). For three-fiber, dual-stimulation experiments, three custom-made fiber-optic cannulae were implanted in the left hemisphere, one above the rCeA/SI (AP 0.60 mm, ML – 2.50 mm, DV 4.40 mm), one above the cCeA (head inclined at a 10° angle; AP 2.15 mm, ML – 3.30 mm, DV 4.10 mm), and one above the VPMpc (AP 1.95 mm, ML – 1.00 mm, DV 3.80 mm). For JAWS-photoinhibition experiments, fiber placement was same for PBN, VPMpc and SI; fibers for CeA were placed at AP 1.10 mm, ML ±3.00 mm, DV 3.85 mm. For all experimental mice, fiber-optic cannulae were affixed to the skull with C and B Metabond (Parkell) and dental acrylic. Mice were

allowed to recover for three wk before the start of behavioral tests. For collateralization-tracing experiments rAAV2-retro Flp virus was injected (0.48 µl unilaterally) into the VPMpc (AP 1.92 mm, ML ±1.00 mm, DV 3.85 mm), PSTN (AP −1.90 mm, ML ±1.50 mm, DV 4.70 mm), CeA (AP 1.10 mm, ML ±3.10 mm, DV 4.10 mm), or ovBNST (AP +0.20 mm, ML ±1.00 mm, DV 4.00 mm) and INTRSECT virus (0.35 µl unilaterally) was injected into the PBN. Tracing mice were sacrificed 4 weeks after virus injection.

## Photostimulation and inhibition

### ChR2

After recovery from surgery, mice were acclimated to dummy cables attached to the implanted fiber-optic cannulas. For behavioral and autonomic studies, bilateral branching fiber-optic cables (200 µm diameter, Doric Lenses) were attached to the head of each mouse before experimentation. Light-pulse trains (10 ms) were delivered at 15 Hz, or 30 Hz as described below. Stimulation paradigms were programmed using a Master8 (AMPI) pulse stimulator that controlled a blue-light laser (473 nm; LaserGlow). The power of light exiting each side of the branching fiberoptic cable was adjusted to 15 ± 0.5 mW. *ChrimsonR* – Same as above, except stimulation was kept to 30 Hz, and the pulse stimulator controlled a red-light laser (660 nm; LaserGlow). The power of light exiting the single fiberoptic (for single-projection terminal stimulation) was adjusted to 5, 12, or 20 mW as described below. For dual-projection terminal stimulation, the light exiting each side of the branching fiberoptic cable was adjusted to 12 ± 0.5 mW. *JAWS* – acclimation same as above, except light was delivered (634 nm, Shanghai Lasers) as 2 s on 1 s ramp 1 s off for continuous inhibition during behavior (e.g. hot-plate test, RTPP), or 3.5 s on 1 s ramp beginning 0.5 s before each 2 s foot shock during foot-shock conditioning. The power of light exiting each side of the branching fiberoptic cable was adjusted to 8 ± 0.5 mW.

## Criteria for exclusion from analysis

Mice were excluded from individual test data if (1) they became immobilized due to tangled fiber-optic patch cords during the behavioral tests, (2) they escaped the arena during photostimulation, or (3) there was limited error-free data collected in pulse-oximeter physiological measurements (this only occurred with respiratory measures). Mice were excluded from all analysis if post-hoc histological examination revealed that viral expression was weak or unilateral, or that fiber-optic cannulae were not appropriately targeted over the projection-site of interest. Locations of fiber tips for all animals that passed the expression and placement criteria are summarized in Figure S3. There was also progressive dropout due to headcap loss requiring animal sacrifice during the study; all data were included up to that point pending histological analysis.

## Slice electrophysiology

Mice were anesthetized with Euthasol (0.2 ml, i.p.) and intracardially perfused with 4–6°C cutting solution containing (in mM): 92 N-methyl-D-glucamine, 2.5 KCl, 1.25 NaH$_2$PO$_4$, 30 NaHCO$_3$, 20 HEPES, 25 D-glucose, two thiourea, 5 Na-ascorbate, 3 Na-pyruvate, 0.5 CaCl$_2$, 10 MgSO$_4$. Coronal slices (300 µm) were cut with a vibratome (Leica VT1200) and kept in the same cutting solution at 33° C for 12 min. Slices were transferred to a 25°C recovery solution containing (in mM): 124 NaCl, 2.5 KCl, 1.25 NaH$_2$PO$_4$, 24 NaHCO$_3$, 5 HEPES, 13 D-glucose, 2 CaCl$_2$, 2 MgSO$_4$. Recordings were made in artificial cerebral spinal fluid (aCSF) containing (in mM) 126 NaCl, 2.5 KCl, 1.2 NaH$_2$PO$_4$, 26 NaHCO$_3$, 11 D-glucose, 2.4 CaCl$_2$, 1.2 MgCl$_2$ continuously perfused at 33°C. All solutions were continuously bubbled with 95%:5% O$_2$:CO$_2$ (pH 7.3–7.4, 300–310 mOsm). Patch-clamp recordings were obtained with a MultiClamp 700B amplifier (Molecular Devices) and filtered at 2 kHz.

### JAWS photoinhibition

CGRP[PBN] neurons expressing AAV1-DIO-JAWS-GFP were identified via epifluorescence and action potentials were recorded in current clamp with patch electrodes (3–5 MΩ) containing (in mM): 135 K-gluconate, 10 HEPES, 4 KCl, 4 Mg-ATP, 0.3 NA-GTP (pH 7.35, 280 mOsm). To assess the effects of CGRP terminal inhibition, excitatory-post synaptic currents (EPSCs) were recorded in voltage clamp at −70 mV from neurons in the CeA surrounded by JAWS:GFP-positive fibers. Patch electrodes (3–5 MΩ) contained (in mM): 117 Cs- MeSO$_3$, 20 HEPES, 0.4 EGTA, 2.8 NaCl, 5 TEA, 4.92 Mg-

ATP, 0.47 Na-GTP (pH 7.35, 280 mOsm). Red light (634 nm, Shanghai Laser) was delivered with a fiber optic placed in the bath above the slice (3 s for action potential recordings and 30 s for EPSCs with 1 s ramp down). EPSCs were analyzed with an automated detection protocol in Mini Analysis Program v.6.0.7 (Synaptosoft) software and manually checked for accuracy.

### Postsynaptic EPSCs
To verify CGRP connectivity to post-synaptic neurons, light-evoked EPSCs were recorded from cells surrounded by ChR2:YFP-positive fibers in each downstream site. Neurons were held in voltage clamp at −70 mV and EPSCs were evoked by 10 ms pulses of blue light delivered through the objective via a 470 nm LED (ThorLabs). Events were analyzed in Clampfit v.11.0.3 (Molecular Devices).

## Behavioral measures
### Order of experiments
Mice were acclimated to handling and attachment of fiber-optic patch cords for one wk, followed by auditory fear conditioning, elevated-plus-maze test, RTPP, unconditioned freezing responses to stimulation in open field, hot-plate test, tail-flick latency test, tail-skin temperature test, autonomic measurements. All replicates were biological (test repetition in biologically distinct samples), not technical (test repetition in same biological sample). Not all cohorts of mice were exposed to all experimental tests – there were biological replicates of mice for PBN photostimulation, and cCeA, SI, and ovBNST terminal photostimulation. The second groups were added for auditory fear conditioning (n = 3.1 (ChR2, YFP) SI only), unconditioned freezing (n = 3.1 ovBNST and SI), and EPM behavioral data (ovBNST and SI), and for the PBN only, plethysmography measurements of respiratory rate (n = 6). Some early groups of PBN stimulation were only tested for unconditioned freezing responses (n = 3). Other variances in group numbers are due to exclusion from individual tests due to adverse events during the test or drop-out due to damaged fiber-optic cannulae (see exclusion criteria, above).

### Auditory fear conditioning
The fear-conditioning chamber was a square arena (25 × 25 cm) with metal walls, two speakers attached on opposite walls, and a metal grid floor that consisted of a circuit board that delivers electrical shock (Coulbourn Instruments). A USB camera was connected to the personal computer and video tracking software (EthoVision XT 10, Noldus Technology) controlled the circuit and recorded the data. Day 1: Mice were attached to fiberoptic patch cords and allowed to habituate for 5 min in their home cage prior to introduction to conditioning context. After free exploration of the context for 1 min, 6 CS tones (tone: 10 kHz 20 s, 60 dB) were played at random intervals, with an average inter-trial interval (ITI) of 2 min. Day 2: Mice were attached and allowed to explore for 1 min; then 6 CS presentations (20 s, 60 dB, 10 kHz) were played at random intervals, with an average ITI of 2 min and each co-terminated with a 2 s light train (30 Hz, 15 mW). Following the sixth CS-US pairing, mice remained in the context for 1 min before being returned to their home cage. Day 3: Mice were attached to fiberoptic patch cords and habituated as before, but then they were placed in a novel context (25 × 25 cm, semitransparent plexiglass). After 2 min of free exploration, one tone CS was played. All the trials were recorded by a USB camera attached to the personal computer and the time spent freezing (during the tone), defined as immobility up until any movement of the head or body, was manually scored with a stopwatch (experimenter was blind to treatments). *With photoinhibition* – same as above, except 2 s light train was replaced with a 2 s 0.5-mA footshock with red light delivery for photoinhibition (8 mW, 3.5 s on, 1 s ramp off, turned on 0.5 s before the shock and ending 2.5 s later).

### Elevated-plus maze (EPM)
The custom-made EPM consisted of two sets of crossed arms (two arms enclosed by 30 cm tall transparent plexiglass, two arms open), each 50 cm long and 8 cm wide, set 65 cm above floor. Mice were attached to fiber optic patch cords and allowed to habituate for 10 min in their home cage prior to introduction to the EPM. Mice were placed in an open arm, 10 cm out, facing the center, with the fiber optic patchcord (4 m long) secured to the ceiling above the center of the maze. Mice were allowed to explore the arena for 10 min with optogenetic stimulation (15 Hz, 2 s on/2 s off).

The sessions were recorded by an USB camera attached to a personal computer and were analyzed using video-tracking software (EthoVision XT 10).

### Real-time place preference (RTPP)

The testing apparatus was a custom-made, three-chambered box (two 18 × 20 cm chambers joined by a 10 × 20 cm start chamber) constructed of opaque black plexiglass with a cement floor. One chamber had walls with vertical pink stripes (2 cm wide), the other had horizontal pink stripes (2 cm wide), and the start chamber had no stripes. Mice were attached to fiber-optic patch cords and allowed to habituate for 10 min in their home cage prior to introduction to the test box. Mice were then introduced to the start chamber and allowed to explore freely during the 15 min trial. One chamber of the box was assigned as the light-paired side. Each time the mouse crossed into the stimulation chamber it received 15 Hz photostimulation or 2 s on 1 s ramp 1 s off trains of photoinhibition until it left the light-paired side. Behavioral data were recorded via an USB camera interfaced with EthoVision software (Noldus Information Technologies).

### Stimulation in open field

Mice were attached to fiber-optic patch cords and allowed to habituate for 5 min in their home cage prior to placement in the arena (40 × 40 cm, white plexiglass walls). One minute after introduction to the arena it received 30 s photostimulation (30 Hz, 15 mW) three times with 60 s inter-stimulation intervals. The sessions were recorded with an USB camera attached to a personal computer and the time spent freezing, defined as immobility up until any movement of the head or body, was manually scored with a stopwatch (experimenter was blind to treatments). Locomotor data was collected using video-tracking software (EthoVision XT 10).

### Hot-plate test – photostimulation

Mice were attached to fiber-optic patch cords and allowed to habituate for 10 min in their home cage prior to stimulation. Following habituation, mice received photostimulation (30 Hz, 8 s on/5 s off, 15 mW) for 7 min prior to exposure to the hot plate. After terminating photostimulation to prevent freezing interfering with responses to heat, mice were placed on the pre-heated aluminum plate (15 × 15 cm, set to 52°C) of the Hot/Cold Plate Analgesia Meter (Coulbourn Instruments). The transparent Plexiglas chamber (15 × 15×20 cm) prevented the mouse from escaping. The latency of the responses to the heat (paw lick, or jump) was measured manually by the experimenter with a stopwatch during the 60 s trials. Trials were recorded with a USB camera attached to a personal computer, and later jump number (jump counted when all four limbs left floor) and the latency to the first jump were manually scored with a stopwatch. *Photoinhibition* – same as above, except the hot plate was set to 57°C, and photoinhibition (2 s on 1 s ramp 1 s off throughout trial) began immediately prior to placing the subject on the plate. Trial was terminated at 30 s.

### Tail-flick-latency test

Mice were attached to fiber-optic patch cords and allowed to habituate for 10 min in their home cage prior to stimulation. Following habituation, mice received photostimulation (30 Hz, 8 s on/5 s off, 15 mW) for 7 min. After ending photostimulation (to prevent freezing interfering with tail-flick reflex), the mouse was restrained within a thick cloth, with only its tail protruding, and its tail was partially submerged (1/2 of its length) into water maintained at 52.5°C (±0.2°C). The tail-flick latency in response to heat was manually scored with a stopwatch. Trials were cut-off at 15 s if no response occurred.

### ChrimsonR or ChR2, single-fiber, freezing responses

Mice were attached to a single, fiber-optic patch cord and allowed to habituate in their home cage for 5 min. After habituation, they were placed into an empty, clean, standard cage, and allowed to explore for 2 min, then they received 10 s photostimulation (30 Hz, 5, 12, or 20 mW). The sessions were recorded with a USB camera attached to a personal computer and the time spent freezing, defined as immobility up until any movement of the head or body, was manually scored with a stopwatch (experimenter was blind to treatments).

## Autonomic measurements

### Tail-skin temperature measurements

Mice were attached to fiber-optic patch cords and allowed to habituate for 10 min in their home cage prior to stimulation. Following habituation, a baseline thermal image of the tail was taken using an infrared camera (FLIR E4; FLIR Instruments). After 2 min of photostimulation (30 Hz, 8 s on/5 s off), a second thermal image was taken. Images were uploaded and analyzed using the software provided (FLIR Tools). Temperature data were taken from 1/3 of length below the base of the tail.

### Pulse-oximeter measurements

Mice were habituated to dummy collar sensors (Starr Life Sciences) for 12 hr overnight prior to secondary habituation to collar sensors and attached cables (Starr Life Sciences). After a full day of habituation, hair was removed from the sensor areas (circumference of neck) to allow trans-dermal infrared penetration, and mice were switched to dummy collar sensors overnight. The next morning, collar sensors and attached cables were placed on the mice, which habituated for at least 30 min prior to patch-cord attachment. Mice were then attached to fiber-optic patch cords and returned to their home cage and allowed to habituate for 1–2 hr, until heart rate and respiration became stable. The collar sensors were attached to a pulse oximeter (MouseOx Plus, Starr Life Sciences) via 3 m cables, and the pulse oximeter was attached to a personal computer via USB. Eventually 5 min of baseline was recorded using the software (Conscious Software Module, Starr Life Sciences), after which the mouse received 3 min of photostimulation (15 or 30 Hz) followed by 1 min of post-stimulation measurements. Recordings were exported and analyzed in Excel.

### Plethysmography measurements

A new cohort of mice (n = 6) was generated to stimulate $CGRP^{PBN}$ neuron somata to measure respiration rate by plethysmography because pulse-oximeter measurements were unable to resolve respiratory rate during somata stimulation. Animals were briefly anesthetized, attached to a bilateral fiber optic patch cord with a rotary joint, and placed in a barometric chamber supplied with room air (21% O2, 200 ml/min). The chamber was sealed for each recording session, which consisted of five recording blocks, 30 s each, centered around 10 s of stimulation (30 Hz) during which the pressure difference was measured between the experimental and reference chamber with a differential pressure transducer. Signals were amplified, digitized, and low-pass filtered (0.1 Hz). Data were collected and analyzed using pCLAMP 9.0 software (Molecular Devices).

## Histology

### Stimulation prior to euthanasia

Mice were attached to fiber-optic patch cords and allowed to habituate for 10 min in their home cage, after which they received 25 min of photostimulation (30 Hz, 3 s on/2 s off). Then they were detached from the patch cords and left in their home cage for 70 min until euthanasia.

### Histology and microscopy

Mice were anesthetized with Beuthansia (0.2 ml, i.p.; Merck) and perfused transcardially with PBS followed by 4% PFA in PBS. Brains were post-fixed overnight in 4% PFA at 4°C, cryoprotected in 30% sucrose, frozen in OCT compound (ThermoFisher), and stored at −80°C. Coronal sections (30 μm) were cut on a cryostat (Leica Microsystems) and collected in cold PBS. For immunohistochemistry experiments, sections were washed three times in PBS with 0.2% Triton X-100 (PBST) for 5 min and incubated in blocking solution (3% normal donkey serum in PBST) for 1 hr at room temperature. Sections were incubated overnight at 4°C in PBST with primary antibodies including: rabbit anti-c-Fos (1:2000, Abcam, ab190289), goat anti-c-Fos (1:500, Santa Cruz Biotechnology, sc-52), chicken-anti-GFP (1:10000, Abcam, ab13970). After three washes in PBS, sections were incubated for 1 hr in PBS with secondary antibodies: Alexa Fluor 488 donkey anti-chicken, Alexa Fluor Cy5 donkey anti-chicken, Alexa Fluor 594 donkey anti-mouse, Cy5 donkey anti-goat, and/or Cy5 donkey anti-rabbit (1:500, Jackson ImmunoResearch). Tissue was washed three times in PBS, mounted onto glass slides, and coverslipped with Fluoromount-G (Southern Biotech). Fluorescent images were acquired using a

confocal microscope. All digital images were processed in the same way between experimental conditions to avoid artificial manipulation between different datasets.

## Collaterals tracing quantification

Coronal sections (30 μm) were collected in 180 μm series and stained for YFP (chicken-anti-GFP; Alexa Fluor Cy5 donkey anti-chicken). Fluorescent images (20X magnification) of each projection target were acquired using a confocal microscope, with the same settings used across all samples and subjects. Across subjects, on average 6 PBN images, 3 VPMpc images, 5 PSTN images, 6 CeA images, 5 SI images, four ovBNST images, and 8 IC images were collected from each brain. Area-specific, pixel-intensity measures for each image/projection target were analyzed in Image-J. Background was subtracted for each image using the average fluorescence from a region of the image outside the projection target analyzed. Pixel-intensity values were summed across individual sections to give the total for each projection target. This value was normalized to either (1) the total pixel intensity values for all areas within subject for % total projection strength, a measure of the contribution of the individual projection to the total projection distribution for the subject or (2) the area-specific pixel intensity in control mice expressing tracer in all CGRP neurons for % maximal pixel intensity, a measure of the projection strength relative to the control condition.

## Collateralization coefficient

To calculate the relative importance of a target structure for contributing the signal in other projection regions we calculated the difference between the normalized $Flp_{on}$ and $Flp_{off}$ fluorescent signal conditions within each downstream region. This value, which ranges between $-1$ and $+1$, equals 0 when fluorescence in the downstream structure is equal when driven only by target-projectors and when only target-projectors are excluded. We set this 0 value to equal 50% by making 50% the y-intercept, then scaled by 50% so that when values are at their maximal (at either $+1$ or $-1$), the value reaches either 0 or 100%.

$$CC_a = \frac{\left([F_a]_{FlpON_b} - [F_a]_{FlpOFF_b}\right)}{[F_a]_{YFP}} \times 50\% + 50\%$$

Here, the target structure of interest is *b*, and the collateralization coefficient is being calculated for its relationship with area *a*. Each target structure (i.e. the VPmpc, PSTN, CeA, ovBNST) will have a number of collateralization coefficients for its relationship with other downstream structures (n = 6 structures $-1$ target = 5). We then averaged across subjects to get the mean collateralization coefficient for each target-area combination and compared the distribution of these values across target areas to assess their relative collateralization tendencies.

## Quantification and statistical analysis

All data were analyzed using Prism 8.0 (GraphPad Software) as described in Supplemental Information. In brief, no tests were used to determine normality of data distributions or to pre-determine sample size; sample size was chosen based on past experience with expected effect sizes. Within-subject data was analyzed using two-sided, paired t-tests; across subject analysis was done with a combination of Welch's t tests (unpaired, correction for no assumption of equal standard deviations), ordinary one-way ANOVA (with Tukey's or Dunnettt's correction for multiple comparisons), and ordinary or repeated measure two-way ANOVAs (with Sidak's correction for multiple comparisons). For two-way ANOVAs, p-value for Treatment (i.e. ChR2 vs YFP)<0.05 is indicated to the right of each graph, and post-hoc row analyses' p-values<0.05 are listed above individual data points.

## Acknowledgements

We thank G Stuber, L Zweifel, C Campos, A Dhaka, B Land, and J Kim for insightful discussions, B Land for guidance with analgesia assays,, J Ramirez for access to plethysmography chambers, G Stuber for use of electrophysiology rig, and J Allen for production of AAV viruses. AJB was supported by a National Institutes of Health T32 Graduate Training grant (T32NS099578). RDP received support from a National Institutes of Health grant (R01-DA24908).

## Additional information

### Competing interests
Richard D Palmiter: Reviewing editor, *eLife*. The other authors declare that no competing interests exist.

### Funding

| Funder | Grant reference number | Author |
| --- | --- | --- |
| National Institutes of Health | T32NS099578 | Anna J Bowen |
| National Institutes of Health | R01-DA24908 | Richard D Palmiter |

The funders had no role in study design, data collection and interpretation, or the decision to submit the work for publication.

### Author contributions
Anna J Bowen, Conceptualization, Data curation, Formal analysis, Funding acquisition, Validation, Investigation, Visualization, Methodology, Writing - original draft; Jane Y Chen, Nathan A Baertsch, Data curation, Formal analysis, Investigation, Methodology, Writing - review and editing; Y Waterlily Huang, Sekun Park, Investigation, Writing - review and editing; Richard D Palmiter, Conceptualization, Supervision, Funding acquisition, Project administration, Writing - review and editing

### Author ORCIDs
Anna J Bowen https://orcid.org/0000-0002-8911-2572
Jane Y Chen http://orcid.org/0000-0002-3986-8785
Nathan A Baertsch http://orcid.org/0000-0003-1589-5575
Richard D Palmiter https://orcid.org/0000-0001-6587-0582

### Ethics
Animal experimentation: This study was performed in strict accordance with the recommendations in the Guide for the Care and Use of Laboratory Animals of the National Institutes of Health. All of the animals were handled according to approved institutional animal care and use committee (IACUC) protocols (#2183-02) of the University of Washington. All surgery was performed under isoflurane anesthesia, and every effort was made to minimize suffering.

### Decision letter and Author response
Decision letter https://doi.org/10.7554/eLife.59799.sa1
Author response https://doi.org/10.7554/eLife.59799.sa2

## Additional files

### Supplementary files
- Supplementary file 1. Full statistical information for all data figures.

- Transparent reporting form

### Data availability
All data generated or analysed during this study are included in the manuscript and supporting files.

The following dataset was generated:

| Author(s) | Year | Dataset title | Dataset URL | Database and Identifier |
| --- | --- | --- | --- | --- |
| Palmiter RD | 2020 | Dissociabe control of unconditioned responses and | https://doi.org/10.5061/dryad.rn8pk0p7k | Dryad Digital Repository, 10.5061/ |

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
