## [Decision Letter]

**Acceptance summary:**

Calcitonin gene-related peptide (CGRP) expressing neurons in the brainstem parabrachial nucleus provide an aversive signal to the brain, inducing a variety of fear related behavioral and autonomic responses. How these cells orchestrate these many functions was not known. This paper directly addresses this important question and reveals unique contributions of distinct efferent projections of CGRP neurons to different aspects of threat processing.

**Decision letter after peer review:**

Thank you for submitting your article "Dissociable control of unconditioned responses and associative fear learning by parabrachial CGRP neurons" for consideration by *eLife*. Your article has been reviewed by three peer reviewers, including Joshua Johansen as the Reviewing Editor and Reviewer #1, and the evaluation has been overseen by Christian Büchel as the Senior Editor.

The reviewers have discussed the reviews with one another and the Reviewing Editor has drafted this decision to help you prepare a revised submission.

Summary:

In this study, the authors studied the efferent anatomical organization and functional role of projections from CGRP cells in the parabrachial (PBN) nucleus which are important in aversive learning and defensive behaviors. The authors used intersectional viral-genetic strategies to systematically dissect the axonal collateralizations of parabrachial CGRP neurons, and examined the consequences of stimulating their projections to individual brain regions as well as to combinations of targets. Both behavioral and autonomic responses (heart rate, respiration, skin temperatures) were carefully examined, and the behavioral tests included fear and anxiety-like behaviors, conditioned learning, and real time place preference/avoidance. This study moves us beyond looking at a behavioral circuit as a simple set of serial connections from one region or set of identified cells to another, to an understanding of the varied and interacting roles that projection collaterals of a single set of cells to many brain regions serve. The PBN is an ideal brain region to disentangle the distinct functional roles of projections to different brain regions which execute the various aspects of this "alarm" system. Through comprehensive studies, the authors discovered that discrete projections from CGRP-PBN neurons produce distinct phenotypes (arousal, freezing or aversive learning), and that combinatorial activation of multiple projection pathways results in emergent phenotypes not seen with individual projections. These findings are novel and will be of broad interest for people studying pain and aversive learning and memory. However, there are some concerns regarding the analyses and interpretation of the data which should be addressed.

Essential revisions:

1) Optogenetic terminal stimulation is artificial and the effects of this stimulation could vary depending on a number of factors. For example, in some cases, the ability of optogenetic stimulation to elicit postsynaptic responses is TTX resistant, suggesting action potentials are not necessary and light-activated opening of ChR2s is enough to activate voltage-dependent Ca channels or to trigger exocytosis by direct Ca^2+^ entry through ChR2s. In other cases, it is TTX-sensitive, i.e., requires action potential generation at the axons. As this situation might depend critically on the quantity of ChR2 molecules transported from the soma to the terminals, this may depend on the length of axon fibers from the soma, axonal transport rate and incubation period after viral vector infection. As the axon length and transport rate should differ between the axons targeting distinct regions, the synaptic efficacy should be crucially different between different target regions, particularly when light-stimulated at the level of terminals. Thus differences seen in the strength of synaptic connectivity of CGRP cells with downstream target neurons or even in the effects of optogenetic stimulation on behavior could be partially due to a variety of factors. The authors should discuss this limitation.

2) Related to the first point, it would be interesting to know why the light-evoked EPSC amplitude differs so much between the target regions in a manner independent of the "projection intensity". Were there differences in release probability, failure rate and paired-pulse ratio? What are the figures (5/6, 5/5, etc.) below the traces in Figure 1J? Do the authors have any interpretation regarding the difference in the size of postsynaptic responses and that regarding the discrepancy between the projection intensity and synaptic intensity?

3) The Palmiter lab's work and many other studies have revealed the importance of the PB in pain processing and behavior. In this study, stimulation of PB-CGRP cells and some of the individual projections of these cells produce robust freezing behavior. However, freezing is not a noxious stimulus induced unconditioned response (UR). The UR to noxious stimuli is an activity burst (1,2) and freezing following noxious stimulation is a learned response. The authors mention this in the Discussion, but throughout the rest of the paper they refer to freezing as the UR. So one important question is whether the freezing they see is in fact learned contextual freezing which is induced by (and secondary to) an aversive property of stimulating the pathway under study and not simply a freezing behavior that is directly elicited by the stimulation. If so, this would change the interpretation of the freezing/learning results. If it is in fact not time locked to the stimulation, the learning in some cases may be transient (no sustained long term memory induced, more like short term memory which degrades quickly). To examine this they could look at how temporally locked the freezing response is to the laser onset/offset in finer temporal detail. Supporting the idea that the freezing is not temporally locked, they report in various panels of Figure 7 that the stimulation produces freezing that lasts into the non-stimulation period.

4) In the hot-plate and tail-flick latency assays, the authors carried out the behavioral tests after having light-stimulated distinct targets for 7 min. First, what was the rationale for stimulating for 7 min? Second, why didn't they examine behavioral influences of stimulation immediately after or during the light-stimulation? Third, how did the authors confirm that this setting was optimal to observe the stimulation effect?

5) Given the authors previous work suggesting the importance of the PB-CeA pathway for aversive learning (3), one surprising finding is that the CGRP-PB projection to the CeA is not important for fear learning (as the authors discuss). However, another study (4) showed that stimulation of PB projections to CeA (not genetically defined, cell type specific) is sufficient to produce escape behaviors and fear learning as well as conditioned place aversion. Together, this suggests that the PB-CGRP cells may only be one component of the PB projection to CeA and that other PBN components/cell types are important for aversive learning. Another possibility is that there are mouse-rat differences as the two papers are from different species. The authors should cite and discuss this prior work and acknowledge these possibilities.

References:

1) Fanselow, 1982

2) Landeira-Fernandez et al., 2006

3) Han et al., 2015

4) Sato et al., 2015

---

## [Author Response]

Essential revisions:1) Optogenetic terminal stimulation is artificial and the effects of this stimulation could vary depending on a number of factors. For example, in some cases, the ability of optogenetic stimulation to elicit postsynaptic responses is TTX resistant, suggesting action potentials are not necessary and light-activated opening of ChR2s is enough to activate voltage-dependent Ca channels or to trigger exocytosis by direct Ca^2+^ entry through ChR2s. In other cases, it is TTX-sensitive, i.e., requires action potential generation at the axons. As this situation might depend critically on the quantity of ChR2 molecules transported from the soma to the terminals, this may depend on the length of axon fibers from the soma, axonal transport rate and incubation period after viral vector infection. As the axon length and transport rate should differ between the axons targeting distinct regions, the synaptic efficacy should be crucially different between different target regions, particularly when light-stimulated at the level of terminals. Thus differences seen in the strength of synaptic connectivity of CGRP cells with downstream target neurons or even in the effects of optogenetic stimulation on behavior could be partially due to a variety of factors. The authors should discuss this limitation.

This is an excellent point – differences in timing of ChR2 transport to terminals across projection targets could indeed lead to changes in observed light-evoked responses and robustness of behavioral or physiological responses that could confound perceived differences in function across the circuit. We considered this possibility when designing our studies, and purposefully waited for virus to express over 4 weeks before beginning behavioral experiments and at least 3 weeks before conducting electrophysiological measurements, in the hopes that since viral expression had become stably and visibly robust at the most distal projection sites that differences in ChR2 transportation would not be the primary driver of cross-area stimulation differences. In support of this being the case, synaptic strength as measured by EPSC amplitude is equal between projection targets that are the most distal and caudal from the PBN – the PSTN and ovBNST, which are approximately 2-mm apart AP. Moreover, differences in ChR2 expression in terminals would likely lead to a difference in activation robustness and hence would drive variation in response amplitude rather than type: for instance, we observed robust freezing behavior from VPMpc stimulation and robust physiological responses with ovBNST stimulation (the most distal projection); it is unlikely that slightly higher levels of ChR2 at terminals in the ovBNST would entirely change the type of response observed, and so cross-areal differences in response types should still be preserved across different ChR2 expression levels. What is more worrisome would be an entire lack of response elicited by stimulation in distal regions. However, nothing except repeated recordings from downstream neurons at different stages of viral expression until EPSC amplitude plateaus at each target-location can truly reveal the time-courses of ChR2 transportation to terminals. We have included a discussion of this limitation in the revised manuscript (subsection “Generation of unconditioned behavioral and physiological responses”).

2) Related to the first point, it would be interesting to know why the light-evoked EPSC amplitude differs so much between the target regions in a manner independent of the "projection intensity". Were there differences in release probability, failure rate and paired-pulse ratio? What are the figures (5/6, 5/5, etc.) below the traces in Figure 1J? Do the authors have any interpretation regarding the difference in the size of postsynaptic responses and that regarding the discrepancy between the projection intensity and synaptic intensity?

We agree that the observed differences in EPSC amplitude across target regions is surprising. We initially thought there must be something affecting viral expression, but found the differences were consistent across expression time and animals. We were able to look at PPR across regions, but with the low number of cells the results were highly variable and did not give rise to anything conclusive. If necessary, we can add new experiments at a later date to examine differences in release probability, failure rate, and PPR across target regions. We suspect that differences in receptor expression or synapse number probably give rise to the differences in EPSC amplitude – especially interesting, the CGRP-receptor, Calcrl, is not expressed in the VPMpc, but is in most other projection targets (Allen brain gene-expression atlas). Our best guess is that the PBN-thalamic projection relies primarily on fast glutamatergic transmission and so may express more AMPA-type receptors, while extended amygdala structures have been shown to richly express Calcrl and other neuropeptide receptors (so-called CGRP^PBN^ neurons express many different neuropeptides in addition to CGRP including Neurotensin, PACAP and Substance-P) and so may rely less prominently on AMPA for excitation (Okutsu et al., 2017). It is also important to clarify that ‘projection intensity’ refers to the total fluorescence, across multiple sections – the CeA, a much longer structure than the VPMpc necessarily has a greater ‘projection intensity’ due to the measure’s cumulative nature. The difference does not necessarily imply that there are more individual cell-to-cell contacts. To clarify this point we changed ‘fiber density’ to ‘cumulative projection strength’, and added discussion about what may potentially give rise to the differences in EPSC amplitude across target regions (subsection “Generation of unconditioned behavioral and physiological responses”).

3) The Palmiter lab's work and many other studies have revealed the importance of the PB in pain processing and behavior. In this study, stimulation of PB-CGRP cells and some of the individual projections of these cells produce robust freezing behavior. However, freezing is not a noxious stimulus induced unconditioned response (UR). The UR to noxious stimuli is an activity burst (1,2) and freezing following noxious stimulation is a learned response. The authors mention this in the Discussion, but throughout the rest of the paper they refer to freezing as the UR. So one important question is whether the freezing they see is in fact learned contextual freezing which is induced by (and secondary to) an aversive property of stimulating the pathway under study and not simply a freezing behavior that is directly elicited by the stimulation. If so, this would change the interpretation of the freezing/learning results. If it is in fact not time locked to the stimulation, the learning in some cases may be transient (no sustained long term memory induced, more like short term memory which degrades quickly). To examine this they could look at how temporally locked the freezing response is to the laser onset/offset in finer temporal detail. Supporting the idea that the freezing is not temporally locked, they report in various panels of Figure 7 that the stimulation produces freezing that lasts into the non-stimulation period.

This is indeed an important consideration, given that we hoped to disentangle ‘unconditioned’ from ‘conditioned’ behaviors with our assays. First, it is important to consider that CGRP^PBN^ neurons have previously been shown to be activated following auditory fear conditioning by CS delivery (Campos et al., 2018) and inactivation of this signal accelerates fear-memory extinction, suggesting that these neurons contribute to the affective signal associated with both US and post-conditioning CS delivery. In this context, it becomes apparent that ascending output from these neurons can both promote escape and freezing behaviors. Clearly, however, the freezing behavior observed following conditioning is still part of a conditioned response that CGRP^PBN^ neuron activity appears to amplify. As yet unpublished work in the Palmiter lab has shown that inactivating CGRP^PBN^ neurons attenuates unconditioned freezing in response to several aversive stimuli, including 20-kHZ USVs and looming stimuli. We believe that CGRP^PBN^ neurons do contribute to freezing behavior as a motor response to potentially dangerous environmental stimuli, a situation which motivated our attempts to discover which downstream populations were best able to recapitulate the robust freezing phenotype generated by directly activating CGRP^PBN^ neurons. To address the important point that transient contextual conditioning could potentially be the primary driver of the observed freezing responses generated by terminal stimulation, we analyzed the latency of freezing responses after beginning and ending stimulation, and compared the relative amount of freezing that occurred during vs. immediately after photostimulation ended. If freezing behavior was generated by transient contextual conditioning, then freezing levels at the beginning and end should, at least at first, be comparable. We found several new interesting results: first, stimulation of terminals in the target regions that best-elicited auditory fear learning and generated freezing behavior (the VPMpc, PSTN, and SI) had the largest differences between freezing behavior across the stimulation and post-stimulation epochs, suggesting that the freezing behavior generated by these structures is not secondary to contextual learning, and that these two associative processes are partially separable at the level of the PBN, although direct analysis of contextual versus auditory fear learning is necessary to confirm this. Moreover, we found that stimulating terminals in the rCeA, part of the nociceptive amygdala, did not promote freezing during stimulation, but caused short-latency freezing bouts following stimulation cessation, reminiscent of UR to contextual freezing response progression caused by noxious stimulus presentation. Another potential explanation for the ‘residual’ freezing behavior we observed is that neuropeptides are released during the stimulation period and could continue to bind receptors downstream following stimulation-cessation, leading to residual activity in downstream sites causing continued, though less robust, generation of freezing behavior. To address these concerns we added a supplement to Figure 3 with new analysis, and included an explicit reference to the issue in the text (subsections “Individual downstream targets of CGRP^PBN^ neurons exert diverse effects on physiology and behavior” and “CGRP^PBN^-neuron downstream targets differentially influence associative learning and affect”).

4) In the hot-plate and tail-flick latency assays, the authors carried out the behavioral tests after having light-stimulated distinct targets for 7 min. First, what was the rationale for stimulating for 7 min? Second, why didn't they examine behavioral influences of stimulation immediately after or during the light-stimulation? Third, how did the authors confirm that this setting was optimal to observe the stimulation effect?

We chose to stimulate for 7 min because we knew we would have to rely on post-synaptic effects of repetitive stimulation to observe alterations in nociceptive processing and we wanted to wait as long as was practicable to lead to the greatest possible effect. This is because activating CGRP^PBN^ neurons leads to such rigid freezing behavior that the tail flick reflex (and obviously paw lick/jumping behavior) is prevented (as mentioned in the Materials and methods), so we knew we had to test without ongoing stimulation. Our experiment was inspired by those examining stress-induced analgesia, where foot shocks are delivered continuously or intermittently for 3 to 30 min to generate profound analgesia (Lewis et al., 1980). We did not confirm that 7 min was optimal. We chose a relatively long stimulation window, tested it, and saw a relatively robust effect and stuck with it.

5) Given the authors previous work suggesting the importance of the PB-CeA pathway for aversive learning (3), one surprising finding is that the CGRP-PB projection to the CeA is not important for fear learning (as the authors discuss). However, another study (4) showed that stimulation of PB projections to CeA (not genetically defined, cell type specific) is sufficient to produce escape behaviors and fear learning as well as conditioned place aversion. Together, this suggests that the PB-CGRP cells may only be one component of the PB projection to CeA and that other PBN components/cell types are important for aversive learning. Another possibility is that there are mouse-rat differences as the two papers are from different species. The authors should cite and discuss this prior work and acknowledge these possibilities.

This is an important consideration, given our finding’s differences from past work examining PBN to CeA projections’ involvement in aversive learning. We agree that it is highly likely that there are other populations within the PBN that receive nociceptive input and may contribute to fear learning via connections to the CeA. A clear example of this possibility is neurons in the dorsal lateral PBN which are now suggested to be the primary receivers of spinoparabrachial input (Chiang et al., 2020), a subset of which projects to the external lateral PBN, where CGRP^PBN^ neurons reside. Indeed, it is possible that CGRP-receptor neurons rely on input from the dlPBN rather than CGRP^PBN^ neurons for their contribution to aversive learning (Han et al., 2015). We have discussed these possibilities (subsection “Associative fear learning”), and are grateful to the reviewer for their helpful suggestions.

References:1) Fanselow, 19822) Landeira-Fernandez et al., 20063) Han et al., 20154) Sato et al., 2015